# A Survey on Federated Fine-Tuning of Large Language Models

**Yebo Wu**                                                                    *yc37926@um.edu.mo*
*State Key Laboratory of IOTSC, University of Macau*

**Chunlin Tian**                                                              *yc27402@um.edu.mo*
*State Key Laboratory of IOTSC, University of Macau*

**Jingguang Li**                                                              *mc45005@um.edu.mo*
*State Key Laboratory of IOTSC, University of Macau*

**He Sun**                                                                        *hesun@um.edu.mo*
*State Key Laboratory of IOTSC, University of Macau*

**Kahou Tam**                                                                *yc37436@um.edu.mo*
*State Key Laboratory of IOTSC, University of Macau*

**Zhanting Zhou**                                                    *ztzhou@std.uestc.edu.cn*
*University of Electronic Science and Technology of China*

**Haicheng Liao**                                                           *yc27979@um.edu.mo*
*State Key Laboratory of IOTSC, University of Macau*

**Jing Xiong**                                                      *junexiong@connect.hku.hk*
*The University of Hong Kong*

**Zhijiang Guo***                                              *zhijiangguo@hkust-gz.edu.cn*
*Hong Kong University of Science and Technology (Guangzhou)*

**Li Li***                                                                          *llili@um.edu.mo*
*State Key Laboratory of IOTSC, University of Macau*

**Chengzhong Xu**                                                             *czxu@um.edu.mo*
*State Key Laboratory of IOTSC, University of Macau*

**Reviewed on OpenReview:** *https://openreview.net/forum?id=rnCqbuIWnn*

## Abstract

Large Language Models (LLMs) have demonstrated impressive success across various tasks. Integrating LLMs with Federated Learning (FL), a paradigm known as FedLLM, offers a promising avenue for collaborative model adaptation while preserving data privacy. This survey provides a systematic and comprehensive review of FedLLM. We begin by tracing the historical development of both LLMs and FL, summarizing relevant prior research to set the context. Subsequently, we delve into an in-depth analysis of the fundamental challenges inherent in deploying FedLLM. Addressing these challenges often requires efficient adaptation strategies; therefore, we conduct an extensive examination of existing Parameter-Efficient Fine-tuning (PEFT) methods and explore their applicability within the FL framework. To rigorously evaluate the performance of FedLLM, we undertake a thorough review of existing fine-tuning datasets and evaluation benchmarks. Furthermore, we discuss FedLLM's diverse

---

*   Corresponding author.

real-world applications across multiple domains. Finally, we identify critical open challenges and outline promising research directions to foster future advancements in FedLLM. This survey aims to serve as a foundational resource for researchers and practitioners, offering valuable insights into the rapidly evolving landscape of federated fine-tuning for LLMs. It also establishes a roadmap for future innovations in privacy-preserving AI. We actively maintain a GitHub repo to track cutting-edge advancements in this field.

# 1 Introduction

Large Language Models (LLMs), such as GPT-4o (OpenAI, 2024), DeepSeek-R1 (Guo et al., 2025), and Qwen3 (Yang et al., 2025) have exhibited extraordinary proficiency across a spectrum of downstream tasks (Xu et al., 2024d; 2025). These LLMs, distinguished by their ability to capture complex semantic knowledge, have established new performance benchmarks in computational linguistics (Xiong et al., 2025; 2024; Xu et al., 2026b). However, despite their impressive capabilities, LLMs cannot be directly deployed for specific downstream tasks without appropriate adaptation (Hu et al., 2021). Furthermore, training LLMs directly on downstream task datasets presents substantial challenges. The massive scale of model parameters leads to significant computational overhead (Tian et al., 2025), while the scarcity of task-specific data constrains effective model training and increases the risk of overfitting. For example, training LLaMA2-65B involves processing approximately 1.4 trillion tokens, requiring 21 days of computation on 2,048 NVIDIA A100 GPUs (Touvron et al., 2023a). Consequently, fine-tuning pre-trained LLMs has become the dominant paradigm (Dodge et al., 2020), enabling more efficient adaptation of LLMs to specific tasks while preserving their foundational knowledge acquired during pre-training.

The current mainstream LLM fine-tuning paradigms can be categorized into three approaches: 1) **Centralized Fine-Tuning** (as shown in Figure 1(a)): This approach aggregates local datasets from all data owners (clients) and uploads them to a central server for fine-tuning (Zhang et al., 2023e; Ding et al., 2023b). Despite its effectiveness, this approach raises significant privacy concerns (Wang et al., 2023b; Ye et al., 2024a; Tam et al., 2024b) and is often impractical in real-world scenarios due to legal restrictions (e.g., GDPR (Voigt & Von dem Bussche, 2017)), which limit the centralization of sensitive personal data. 2) **Local Fine-Tuning** (as shown in Figure 1(b)): In this paradigm, each data owner fine-tunes the LLM locally using their private dataset. While this approach preserves data privacy, the limited size and diversity of local datasets often result in suboptimal model performance. For instance, models refined through local fine-tuning demonstrate a substantial performance degradation of up to 7% on the MMLU benchmark (Hendrycks et al., 2020) when compared to federated fine-tuning (Ye et al., 2024b). 3) **Federated Fine-Tuning** (as shown in Figure 1(c)): This approach enables collaborative model improvement while preserving data privacy by allowing clients to train the model locally and only sharing model updates with the central server (Li et al., 2025; 2023c). The server aggregates these updates to construct a global model, which is subsequently redistributed to clients for further refinement. This method addresses both the privacy concerns of centralized fine-tuning and the limited data diversity issues of local fine-tuning, making it a promising paradigm for adapting LLMs to specific downstream tasks (Xu et al., 2023e).

Despite these benefits, federated fine-tuning encounters several unique challenges, which significantly limit the effective deployment of FedLLM in real-world scenarios: 1) **Communication Overhead**: LLMs typically contain billions of parameters, such as LLaMA2-7B. Therefore, uploading these massive model parameters in each training round incurs substantial communication overhead, resulting in severe communication latency and excessive bandwidth requirements (Li et al., 2022a). 2) **Data Heterogeneity**: Data across participating clients exhibits substantial variation in both quality and statistical distribution (Ning et al., 2024). This Non-IID (Non-Independent and Identically Distributed) nature of federated data can introduce significant biases into model updates, leading to weight divergence, slower convergence rates, and ultimately compromised model performance (Fu et al., 2024a; Tian et al., 2024c). 3) **Memory Wall**: Participating clients, especially edge devices, generally possess limited available memory resources (Wu et al., 2024h;i; Zhan et al., 2024; Li et al., 2023g; Wu et al., 2025c), which insufficiently support memory-intensive LLM fine-tuning. This memory wall fundamentally limits clients' effective participation in the federated fine-tuning process, preventing them from contributing valuable data to the global model and ultimately compromising model

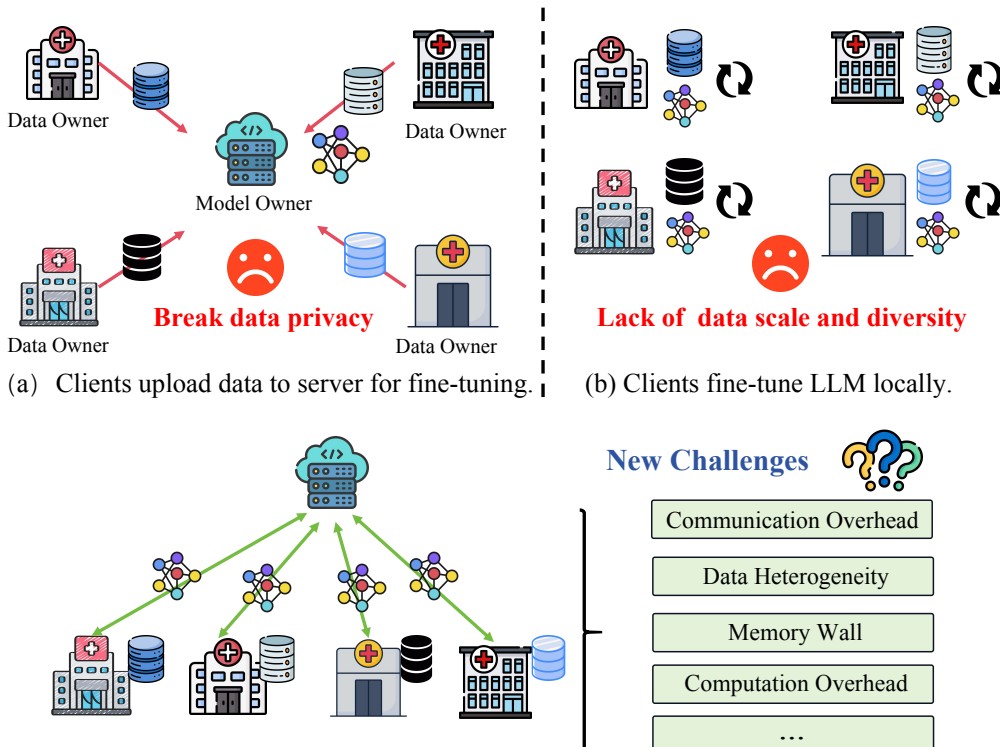

(a) Clients upload data to server for fine-tuning.

(b) Clients fine-tune LLM locally.

(c) Federated fine-tuning of LLMs and its new challenges.

Figure 1: Illustration of three LLM fine-tuning paradigms: (a) **Centralized Fine-tuning**, where data is aggregated at a central server; (b) **Local Fine-tuning**, where models are trained independently on private datasets; and (c) **Federated Fine-tuning**, where data remains local, and model updates are aggregated by a central server to create a global model.

performance. 4) **Computation Overhead**: The hardware processing capabilities of participating clients are often limited (Wang et al., 2019a), making it challenging to meet the high computational demands of fine-tuning LLMs. This computational bottleneck substantially increases local training time and consequently prolongs the overall process. The extended training cycles reduce system efficiency and significantly increase energy consumption on resource-constrained devices, potentially deterring client participation.

To address these challenges, researchers have applied various Parameter-Efficient Fine-Tuning (PEFT) methods to FL, which can be broadly classified into five main categories: LoRA-based tuning (Hu et al., 2021; Tian et al., 2024b; Wu et al., 2025b), prompt-based tuning (Lester et al., 2021), adapter-based tuning (Houlsby et al., 2019), selective-based tuning (Zaken et al., 2021), and other tuning methods (Shin et al., 2023; Chen et al., 2017). The core idea behind these methods is to minimize the number of trainable parameters by focusing on small, task-specific adjustments rather than fine-tuning the entire model. By updating only a subset of parameters or modifying model inputs (e.g., prompts), these approaches significantly reduce the communication overhead, computational burden, and memory usage of model fine-tuning, all while maintaining the performance of LLMs across diverse tasks.

**Prior Surveys.** Despite the continuous development of innovative federated fine-tuning methods, a significant gap remains in the comprehensive evaluation and comparison of these techniques. While existing surveys on FL provide valuable insights, they are typically limited to either traditional small-model FL settings or fail to offer a detailed analysis and evaluation benchmark specifically for federated fine-tuning of LLMs. Concurrently, although several surveys on PEFT have been proposed, these works predominantly focus on centralized fine-tuning scenarios, overlooking the unique challenges that arise when adapting such techniques to distributed, privacy-preserving settings. A detailed comparison of existing surveys is summarized in Table 1. To fill this gap, our survey aims to be the *first* to present a systematic examination of federated

Table 1: **Overview of related surveys.** This comparison highlights whether each work addresses data privacy, aligns with the scope of LLMs, emphasizes efficiency, proposes evaluation benchmarks, and discusses applications and future directions.

| Prior Surveys | Privacy | LLM | Efficiency | Benchmark | Application | Future Direction |
|---|---|---|---|---|---|---|
| Xu et al. (2024c) | ✗ | ✓ | ✓ | ✗ | ✓ | ✓ |
| Zhao et al. (2023b) | ✗ | ✓ | ✓ | ✓ | ✓ | ✓ |
| Gao et al. (2024b) | ✗ | ✓ | ✗ | ✓ | ✗ | ✓ |
| Li et al. (2024d) | ✗ | ✓ | ✗ | ✓ | ✓ | ✓ |
| Zheng et al. (2023b) | ✗ | ✓ | ✗ | ✓ | ✓ | ✓ |
| Han et al. (2024) | ✗ | ✓ | ✓ | ✗ | ✓ | ✓ |
| Xin et al. (2024) | ✗ | ✓ | ✓ | ✓ | ✓ | ✓ |
| Huang et al. (2024b) | ✓ | ✗ | ✗ | ✓ | ✗ | ✓ |
| Ye et al. (2023) | ✓ | ✗ | ✗ | ✗ | ✗ | ✓ |
| Jiang et al. (2024b) | ✓ | ✗ | ✗ | ✗ | ✓ | ✗ |
| Yuan et al. (2024a) | ✓ | ✗ | ✗ | ✗ | ✓ | ✓ |
| Chen et al. (2024c) | ✓ | ✗ | ✗ | ✓ | ✓ | ✓ |
| Chai et al. (2024) | ✓ | ✗ | ✗ | ✓ | ✗ | ✓ |
| Feng et al. (2023b) | ✓ | ✗ | ✗ | ✓ | ✓ | ✓ |
| Zhang et al. (2024j) | ✓ | ✗ | ✗ | ✗ | ✗ | ✓ |
| Yao et al. (2024) | ✓ | ✓ | ✓ | ✗ | ✓ | ✓ |
| Li et al. (2024e) | ✓ | ✓ | ✓ | ✗ | ✓ | ✓ |
| Zhuang et al. (2023) | ✓ | ✓ | ✗ | ✗ | ✓ | ✓ |
| Woisetschläger et al. (2024) | ✓ | ✓ | ✓ | ✗ | ✗ | ✓ |
| Ren et al. (2024) | ✓ | ✓ | ✓ | ✗ | ✓ | ✓ |
| **Ours** | ✓ | ✓ | ✓ | ✓ | ✓ | ✓ |

fine-tuning for LLMs, providing a thorough understanding of their evolution, effectiveness, and practical implementation challenges, along with standardized evaluation benchmarks that enable fair comparison across different approaches. This thorough analysis serves as a foundation for researchers and practitioners seeking to navigate the rapidly evolving landscape of federated LLM fine-tuning.

**Contribution.** This paper presents a comprehensive survey on the federated fine-tuning of LLMs. In contrast to existing surveys, the main contributions of this work can be summarized as follows:

1. We provide an exhaustive review of all relevant papers on federated fine-tuning up to date, offering an extensive analysis of the state-of-the-art techniques and their evolution in this field.

2. We conduct a detailed analysis of the key challenges in federated fine-tuning and propose a systematic taxonomy based on different fine-tuning approaches, including LoRA-based, prompt-based, adapter-based, selective-based, and other emerging tuning methods. We further provide an in-depth discussion of the advantages, limitations, and applicability of these methods.

3. We establish a comprehensive evaluation framework for federated fine-tuning of LLMs, encompassing fine-tuning datasets and evaluation benchmarks across diverse domains, while systematically analyzing and discussing diverse real-world application scenarios.

4. Finally, we outline promising research directions in FedLLM, aiming to guide future investigations toward more efficient, scalable, and privacy-preserving solutions that bridge the gap between theoretical advances and practical deployments in resource-constrained federated environments.

Figure 2 illustrates the organizational structure of this survey. Section 2 introduces the relevant background and fundamental concepts of LLMs and federated fine-tuning. Section 3 systematically examines the technical challenges and inherent limitations in the federated fine-tuning of LLMs. In Section 4, we present a comprehensive review of state-of-the-art federated fine-tuning techniques and methodologies. Section 5 presents representative fine-tuning datasets and evaluation benchmarks across various domains, specifically curated to assess the performance of federated fine-tuning in diverse scenarios. Section 6 explores practical applications of FedLLM. Section 7 outlines promising research directions, while Section 8 synthesizes key insights from this survey to inform and guide future research in this rapidly evolving field.

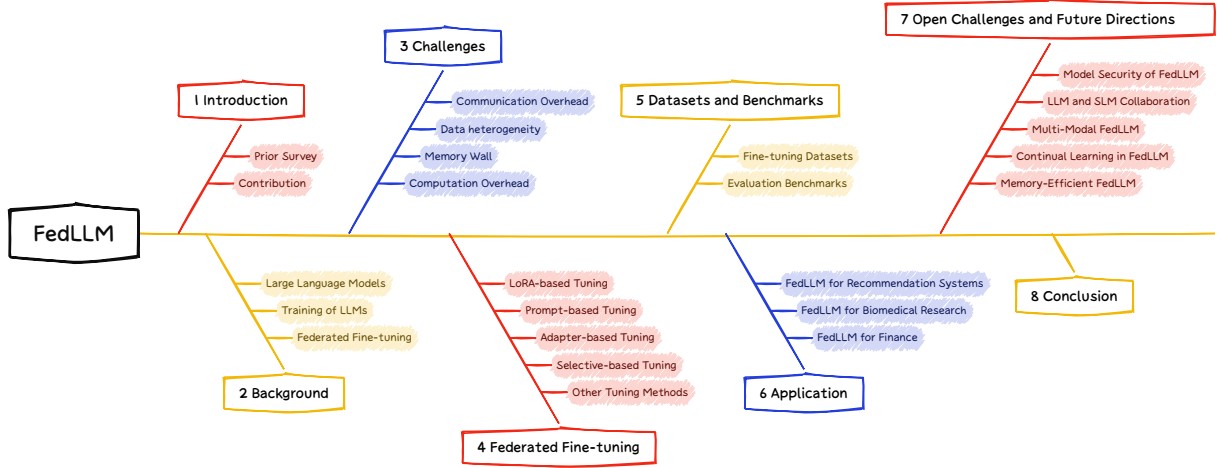

Figure 2: Overall structure of the survey.

## 2 Background

### 2.1 Large Language Models

Large Language Models (LLMs) have demonstrated unprecedented capabilities across a wide range of natural language processing tasks, including machine translation (Wang et al., 2022a), text generation (Yu et al., 2022), sentiment analysis (Wankhade et al., 2022), and question answering (Zhu et al., 2021). Their exceptional performance stems from their remarkable ability to encode complex linguistic patterns, capture long-range contextual dependencies, and learn rich semantic representations (Xiong et al., 2023; Xu et al., 2026a). These capabilities have not only enabled LLMs to achieve state-of-the-art results on a broad spectrum of academic benchmarks, but have also fueled transformative advances in real-world applications such as conversational AI, legal document analysis, medical decision support, and automated content generation.

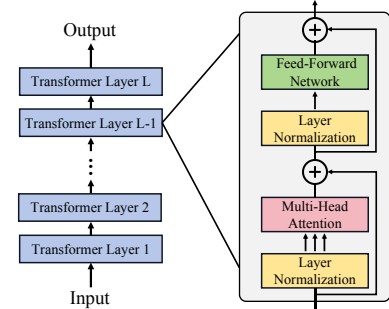

Figure 3: Architecture of LLMs.

Architecturally, modern LLMs are typically constructed by stacking dozens or even hundreds of transformer layers, where each layer incrementally refines the input through deep contextualization and abstraction. For example, LLaMA2-7B comprises 32 transformer layers stacked sequentially to capture hierarchical linguistic features. This deep, layered architecture enables the model to effectively integrate both local and global contextual information over long sequences, which is essential for complex language understanding tasks. Figure 3 illustrates the schematic structure of a prototypical LLM, where transformer layers are arranged in a vertically stacked fashion. Each transformer layer consists of two fundamental components: Multi-Head Attention (MHA) and Feed-Forward Network (FFN). Formally, the input to the $l$-th transformer layer is denoted as $h_{l-1} \in \mathbb{R}^{n \times d}$, where $n$ is the sequence length and $d$ is the hidden dimension of the model. The computational process within the $l$-th layer can be expressed as follows:

$$h_i' = \text{MHA}(\text{LN}(h_{i-1})) + h_{i-1}, \tag{1}$$

$$h_i = \text{FFN}(\text{LN}(h_i')) + h_i' \tag{2}$$

where $\text{LN}(\cdot)$ represents layer normalization, which stabilizes the training dynamics by standardizing the activations, and $h_i'$ denotes the intermediate activations after being processed by the MHA module.

### 2.2 Training of LLMs

The training of LLMs encompasses two distinct stages (Xin et al., 2024): pre-training and fine-tuning, as illustrated in Figure 4. 1) **Pre-training** involves training the model on massive unlabeled text corpora

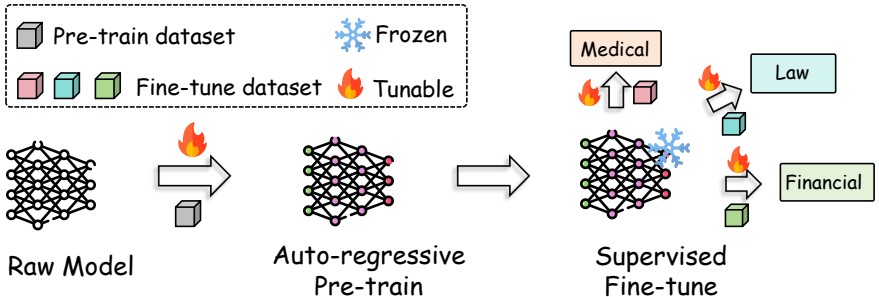

Figure 4: Schematic illustration of the two-stage LLM training process: 1) auto-regressive pre-training on large-scale corpora to develop general linguistic capabilities, followed by 2) supervised fine-tuning to align model outputs with specific task requirements or human preferences.

(billions to trillions of tokens) drawn from diverse sources such as academic papers, websites, and books. This stage generally adopts the auto-regressive modeling (Yang et al., 2019) approach that predicts each token based on its previous context. Through extensive training, LLMs develop a wide range of capabilities, from basic semantic understanding to advanced reasoning across diverse domains. While computationally intensive, this unsupervised learning process builds robust and transferable representations that serve as a powerful foundation for various downstream tasks (Naveed et al., 2023).

2) The function of **fine-tuning** is to adapt the pre-trained model to specific downstream tasks through additional training on task-specific datasets (Ding et al., 2023b). This stage typically utilizes supervised learning to optimize the model's performance for particular applications. While fine-tuning is highly effective at specializing the model's general language understanding for targeted tasks, traditional fine-tuning methods often require centralizing data from various sources on a central server (Huang et al., 2025), which raises significant privacy and security concerns. These concerns have sparked growing interest in privacy-preserving fine-tuning paradigms, which seek to retain the benefits of model specialization while ensuring that sensitive user data remains decentralized and secure throughout the training process.

### 2.3 Federated Fine-Tuning

Federated fine-tuning (Zhang et al., 2024b; Yi et al., 2025) has emerged as a promising paradigm for adapting LLMs to specific downstream tasks while preserving data privacy. Unlike conventional centralized approaches that require sensitive data aggregation at a central server, federated fine-tuning enables distributed clients to adapt LLMs on local private datasets, sharing only model updates with the coordinating server. This privacy-preserving approach aligns well with modern data protection requirements and user expectations. However, the massive scale of LLM parameters and heterogeneous data distributions across clients introduce significant technical challenges. These challenges encompass prohibitive communication bandwidth requirements for transmitting model updates, convergence difficulties when training across heterogeneous data distributions, excessive memory demands that strain client-side resources, and intensive computational overhead that impacts efficiency and energy consumption. To address these challenges, researchers have proposed various parameter-efficient federated fine-tuning approaches, each strategically designed to mitigate resource constraints while maintaining model performance on downstream tasks. These innovative methods can be broadly categorized as follows:

- 1) LoRA-based Tuning (Hu et al., 2021): This methodology leverages the intrinsic low-rank nature of weight updates by decomposing them into low-rank approximation matrices, significantly reducing trainable parameters while preserving model's expressiveness.

- 2) Prompt-based Tuning (Lester et al., 2021): This approach optimizes continuous or discrete prompts in the input space to steer the model's behavior toward specific tasks. By modifying only the prompt embeddings while keeping model weight frozen, it achieves remarkable parameter efficiency in task adaptation.

- 3) Adapter-based Tuning (Houlsby et al., 2019): This strategy incorporates specialized adapter modules between the layers of the pre-trained model. By updating only these compact adapters while freezing the original model parameters, it enables efficient task-specific adaptation with minimal architectural modifications to the base model.

- 4) Selective-based Tuning (Zaken et al., 2021): This approach focuses on selectively fine-tuning specific layers or parameters of the model that are most relevant to the downstream task. Through careful selection, it significantly reduces the resource consumption.

- 5) Other Tuning Methods (Li et al., 2024e): This category encompasses techniques like zeroth-order optimization (Malladi et al., 2023), split learning (Thapa et al., 2022), model compression (Deng et al., 2020), and data selection (Qin et al., 2024), which offer innovative ways to optimize LLM performance with lower resource requirements.

## 3  Challenges

In this section, we provide an in-depth analysis of the challenges encountered in FedLLM, focusing on four key aspects: communication overhead, data heterogeneity, memory constraints, and computation burden.

### 3.1  Communication Overhead

In federated fine-tuning, the learning process necessitates iterative communication between participating clients and the central server, where clients periodically transmit their locally updated model parameters for aggregation (Fu et al., 2022; 2024b;c). This iterative exchange continues until model convergence, inherently introducing substantial communication overhead (Li et al., 2022b; Kou et al., 2025). The challenge is even more pronounced when fine-tuning LLMs, which consist of billions of parameters. To quantify this challenge, Figure 5 presents a comparative analysis of parameter sizes across different models, contrasting traditional models like BERT (Devlin et al., 2019) with the LLaMA series, including TinyLLaMA (Zhang et al., 2024d), LLaMA2-7B, LLaMA2-13B (Touvron et al., 2023b), LLaMA3-3B, and LLaMA3-8B (Dubey et al., 2024). Our analysis reveals that LLaMA models are dramatically larger than BERT, with parameter counts ranging from 10 to 118× greater. This exponential increase in parameter size directly translates to significantly higher data transmission volumes in each communication round, substantially elevating bandwidth requirements and overall communication costs in federated environments.

However, in real-world scenarios, communication bandwidth is often severely constrained. According to a 2023 Cisco report, approximately 30% of edge devices still rely on 2G or 3G networks, which provide bandwidths of less than 10 Mb/s (Wang et al., 2023c). While 5G networks offer speeds more than 50× faster, they are accessible to only about 10% of devices. This significant disparity in network capabilities inevitably results in substantial communication delays, particularly when exchanging large parameter updates. More critically, the duration of each training round is determined by the slowest device in the network—a phenomenon known as the "straggler effect." This means that devices with limited connectivity can dramatically hinder the convergence speed of the federated fine-tuning process. Consequently, minimizing communication overhead becomes essential for effective FedLLM implementation. Efficient management of data transmission can accelerate model convergence while ensuring the practical feasibility of deploy-

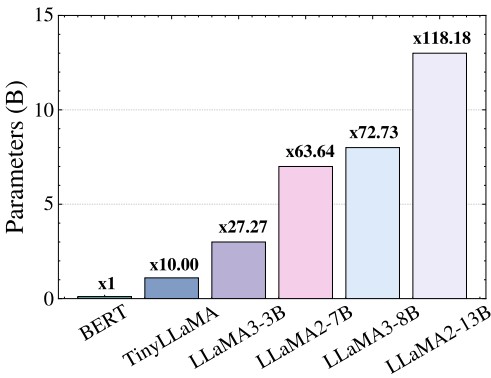

Figure 5: Comparison of model parameters across BERT and LLaMA series models.

ing FedLLM in bandwidth-constrained environments. Without addressing the communication challenge, the theoretical privacy benefits of federated fine-tuning may remain inaccessible to many real-world applications, particularly those involving edge devices or regions with limited network infrastructure.

## 3.2 Data Heterogeneity

Data heterogeneity is a notorious challenge in FL, manifesting in significant variations in data distribution (Tian et al., 2022a), quality (Tam et al., 2023a;b), and quantity (Yi et al., 2022) across clients. Such heterogeneity hinders convergence and degrades the global model's generalization ability, as it must reconcile conflicting updates derived from diverse client populations (Ma et al., 2024a; Wang et al., 2025). To mitigate the adverse effects of data heterogeneity, various strategies have been explored in traditional FL, which can be broadly categorized into four groups: 1) **Regularization-based methods** incorporate additional penalty terms into the local objective to limit model divergence and encourage alignment with the global model (Li et al., 2020); 2) **Aggregation-based methods** modify the server-side aggregation strategy to assign adaptive weights to client updates, reducing the influence of noisy, unreliable, or biased data sources (Wang et al., 2020a;b); 3)

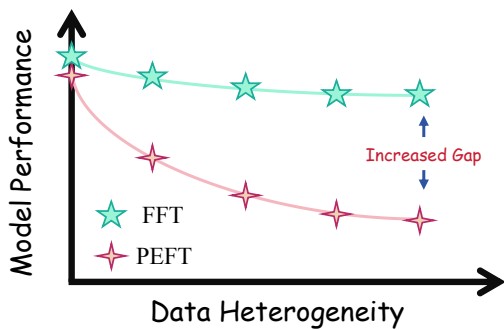

Figure 6: Impact of data heterogeneity on model performance. As the degree of data heterogeneity increases, the performance gap between PEFT and FFT widens.

**Data-sharing methods** introduce small, carefully curated auxiliary datasets that are distributed to clients to promote distributional alignment and reduce inter-client drift (Goetz & Tewari, 2020); 4) **Personalized FL approaches** aim to address data heterogeneity by learning models that capture the unique characteristics of local data. Rather than converging to a single global model, these methods may produce multiple global models that are individually personalized for each client (Kou et al., 2024; Sabah et al., 2024). While these techniques have demonstrated success in traditional FL scenarios, addressing data heterogeneity in FedLLM remains largely underexplored. This challenge is further exacerbated when applying PEFT techniques, as PEFT tends to be more sensitive to distributional shifts and limited data availability. Figure 6 shows that the performance gap between PEFT and full-parameter fine-tuning (FFT) grows wider as data heterogeneity increases, underscoring the need for targeted solutions to improve PEFT robustness in federated settings.

## 3.3 Memory Wall

Memory constraints present a fundamental challenge to the practical deployment of federated fine-tuning (Wu et al., 2025d). During the local fine-tuning process, model parameters, intermediate activations, and gradients must be stored in memory, resulting in substantial memory consumption. However, participating clients, especially edge devices, typically have limited available memory, ranging from 4 to 12 GB (Tian et al., 2024a; Tam et al., 2024a). This limited memory capacity is insufficient to support fine-tuning mainstream LLMs.

To better quantify this challenge, we profile the memory usage during full-parameter fine-tuning for both traditional models (e.g., DistilBERT (Sanh et al., 2020), BERT (Devlin et al., 2019)) and LLaMA-series models (e.g., TinyLLaMA, LLaMA2-7B, LLaMA2-13B). As shown in Figure 7, our results reveal a dramatic disparity in resource requirements between these model families. Fine-tuning LLaMA models demands substantially higher memory resources compared to traditional architectures. Specifically, fine-tuning LLaMA2-7B requires approximately 51.85 GB of GPU memory, which is 7.68× more than BERT (6.75 GB). This requirement escalates further with LLaMA2-13B, which demands 98.56 GB of memory, representing a 28.32× increase over DistilBERT and vastly exceeding the available memory capacity of edge devices. This stark mismatch between the memory demands of fine-tuning LLMs and the hardware limitations of participants creates a **Memory Wall**, a fundamental barrier that severely restricts the feasibility of deploying FedLLM at scale. This memory constraint

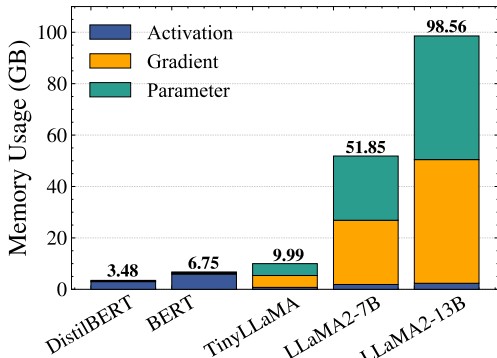

Figure 7: Memory usage breakdown for fine-tuning various models. The analysis is conducted on an NVIDIA H800 with precisions of FP32 and FP16, a batch size of 16, and a maximum sequence length of 512.

prevents a significant proportion of devices from participating in the collaborative learning process, thereby compromising model performance through reduced data diversity and limiting the practical application scope of FedLLM. The memory wall represents not just a technical challenge but a fundamental constraint on democratizing access to advanced AI capabilities through FL.

### 3.4 Computation Overhead

Computational cost presents another major bottleneck in deploying FedLLM (Tian et al., 2022a; Almanifi et al., 2023). The computational demands of fine-tuning LLMs arise from the forward and backward passes during local training iterations, each contributing significantly to the overall processing burden. The sheer scale of these models—characterized by billions of parameters, numerous transformer layers, and complex attention mechanisms—makes them inherently compute-intensive, which can quickly overwhelm devices with limited processing capabilities. Moreover, the iterative nature of fine-tuning, which involves repeated forward and backward passes, compounds the computational load, making it difficult to achieve efficient training on resource-constrained devices. To quantitatively understand this challenge, we profile the computational demands of fine-tuning various models by measuring the floating-point operations (FLOPs) required for a single forward and backward pass with a batch size of 16.

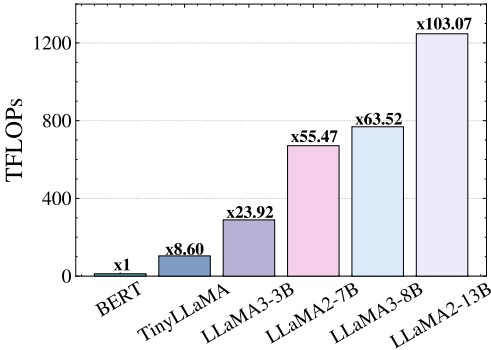

Figure 8: Comparison of FLOPs for a single forward and backward pass across models.

Specifically, we evaluate BERT alongside a suite of LLaMA-based models, including TinyLLaMA, LLaMA3-3B, LLaMA2-7B, LLaMA3-8B, and LLaMA2-13B. As shown in Figure 8, our results reveal a dramatic escalation in computational requirements for LLaMA-series models compared to BERT. For instance, fine-tuning TinyLLaMA incurs $8.60\times$ FLOPs of BERT, while LLaMA2-13B demands a staggering $103.07\times$ more FLOPs. This exponential increase in computational complexity directly results in significantly longer training time, excessive energy consumption on battery-powered devices, and thermal management issues, all of which can degrade hardware performance over time, thereby undermining the feasibility of large-scale, real-world deployment (Ning et al., 2024; Tian et al., 2023). These findings highlight the pressing need for computation-efficient fine-tuning strategies that can effectively accommodate the heterogeneous and resource-constrained nature of participating devices, while ensuring model performance.

## 4 Federated Fine-Tuning

In this section, we introduce various parameter-efficient fine-tuning methods and discuss their applications in FL. Figure 9 provides an overview of representative parameter-efficient federated fine-tuning methods.

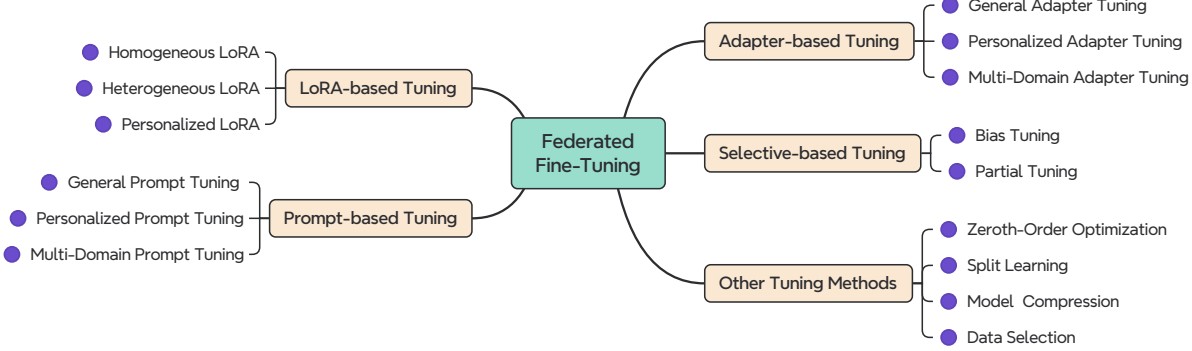

Figure 9: Overview of parameter-efficient federated fine-tuning methods and their corresponding taxonomy.

### 4.1 LoRA-based Tuning

#### 4.1.1 Preliminary

Low-Rank Adaptation (LoRA) (Hu et al., 2021; Tian et al., 2024b) has emerged as a promising approach for efficient fine-tuning of LLMs while maintaining model performance. The core idea of LoRA lies in introducing low-rank matrices into the pre-trained model's weights, allowing for the adaptation of model parameters without altering the original architecture significantly. LoRA is based on the observation that fine-tuning does not require updating the full parameter space; instead, meaningful adaptations can often be represented in a low-dimensional subspace. By applying low-rank decomposition to the weight updates, LoRA drastically reduces the number

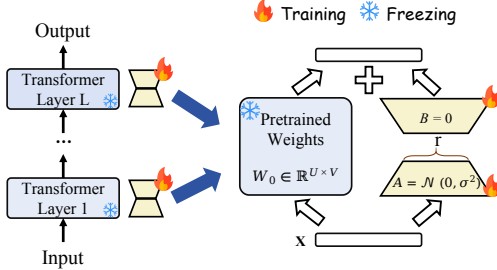

Figure 10: The working principle of LoRA.

of trainable parameters, leading to lower resource consumption. Furthermore, LoRA's modular design allows it to be easily integrated into a variety of model architectures without altering the original model.

Figure 10 illustrates the working principle of LoRA. Specifically, the pre-trained parameter matrix $\mathbf{W_0} \in \mathbb{R}^{U \times V}$ is decomposed into two matrices, $\mathbf{A} \in \mathbb{R}^{r \times V}$ and $\mathbf{B} \in \mathbb{R}^{U \times r}$, where $r \ll \min(U, V)$ denotes the rank controlling the dimensionality of the low-rank subspace. The matrix $\mathbf{A}$ projects the input into a low-dimensional space, and the matrix $\mathbf{B}$ maps it back to the original space. During fine-tuning, only $\mathbf{A}$ and $\mathbf{B}$ are updated, while the original weights $\mathbf{W}_0$ remain frozen. The input data $X$ is processed by both $\mathbf{W_0}$ and $\mathbf{BA}$. The output of $\mathbf{W_0}X$ is the initial prediction generated by the pre-trained model, while the output of $\mathbf{B}\mathbf{A}X$ represents the task-specific adaptation introduced by the low-rank matrices. These two outputs are then added element-wise to produce the final output. This process can be formulated as:

$$h = \mathbf{W_0}X + \mathbf{BA}X \tag{3}$$

By updating only $\mathbf{A}$ and $\mathbf{B}$, LoRA enables efficient task adaptation with minimal resource overhead, effectively capturing task-specific knowledge while preserving the generalization ability of the pre-trained model.

#### 4.1.2 LoRA in Federated Fine-Tuning

In the context of federated fine-tuning, LoRA offers notable advantages in both communication and computation efficiency. Since only the low-rank matrices are updated and transmitted, rather than the full model parameters, this lightweight updating mechanism significantly lowers bandwidth requirements and facilitates faster convergence of the global model, making LoRA particularly well-suited for federated environments. In this paper, we introduce a novel taxonomy of LoRA-based federated fine-tuning methods, categorizing them into three primary types: Homogeneous LoRA, where all clients adopt the same rank; Heterogeneous LoRA, where clients use different ranks based on their resources; and Personalized LoRA, which tailors low-rank adaptations to individual client data distributions. This taxonomy is summarized in Figure 9. In the following sections, we delve into representative methods within each category, analyzing how they address the key challenges identified in Section 3, beyond the inherent benefits brought by LoRA itself.

● **Homogeneous LoRA** refers to scenarios where all clients adopt the same low-rank dimension $r$ for their LoRA modules. This uniform configuration simplifies aggregation and model synchronization across clients. Table 2 summarizes representative methods in this category and the specific challenges they address.

**FedIT** (Zhang et al., 2024b) directly integrates LoRA into the classic FedAvg (McMahan et al., 2017) for instruction tuning. **FedSA-LoRA** (Guo et al., 2024b) identifies that the $\mathbf{A}$ matrices primarily encode general knowledge, while the $\mathbf{B}$ matrices capture client-specific features; thus, it only uploads $\mathbf{A}$ to the server, significantly reducing communication overhead. **FederatedScope-LLM** (Kuang et al., 2023) establishes a comprehensive end-to-end pipeline for federated LLM fine-tuning and proposes offsite-tuning strategies to mitigate both communication and computational costs. **FeDeRA** (Yan et al., 2024) addresses data heterogeneity by initializing LoRA matrices via singular value decomposition on the pre-trained weights. **LoRA-FAIR** (Bian et al., 2024) introduces a correction mechanism on the server to handle aggregation

Table 2: **Homogeneous LoRA in federated fine-tuning.**

| Method | Challenge | | | |
|---|---|---|---|---|
| | Communication | Non-IID | Memory | Computation |
| FedIT (Zhang et al., 2024b) | ✗ | ✗ | ✗ | ✗ |
| FedSA-LoRA (Guo et al., 2024b) | ✓ | ✗ | ✗ | ✗ |
| FederatedScope-LLM (Kuang et al., 2023) | ✓ | ✗ | ✗ | ✓ |
| FeDeRA (Yan et al., 2024) | ✗ | ✓ | ✗ | ✗ |
| LoRA-FAIR (Bian et al., 2024) | ✗ | ✗ | ✗ | ✗ |
| FLASC (Kuo et al., 2024) | ✓ | ✗ | ✗ | ✗ |
| SA-FedLoRA (Yang et al., 2024c) | ✓ | ✓ | ✗ | ✗ |
| SLoRA (Babakniya et al., 2023) | ✗ | ✓ | ✗ | ✗ |
| RoLoRA (Chen et al., 2024d) | ✗ | ✓ | ✗ | ✗ |
| FedPipe (Fang et al., 2024b) | ✓ | ✗ | ✓ | ✓ |
| Lp-FL (Jiang et al., 2023) | ✗ | ✗ | ✗ | ✗ |
| Fed-piLot (Zhang et al., 2024l) | ✗ | ✓ | ✓ | ✗ |
| FedRA (Su et al., 2023) | ✓ | ✗ | ✓ | ✓ |
| FedPruner (Wu et al., 2025c) | ✓ | ✓ | ✓ | ✓ |
| DevFT (Wu et al., 2025b) | ✓ | ✗ | ✓ | ✓ |

Table 3: **Heterogeneous LoRA in federated fine-tuning.**

| Method | Challenge | | | |
|---|---|---|---|---|
| | Communication | Non-IID | Memory | Computation |
| HETLoRA (Cho et al., 2024) | ✓ | ✓ | ✓ | ✓ |
| FLoRA (Wang et al., 2024f) | ✓ | ✗ | ✓ | ✓ |
| FlexLoRA (Bai et al., 2024a) | ✓ | ✗ | ✓ | ✓ |
| LoRA-A$^2$ (Koo et al., 2024) | ✓ | ✓ | ✓ | ✓ |
| Byun & Lee (2024) | ✓ | ✗ | ✓ | ✓ |
| FedHM (Yao et al., 2021) | ✓ | ✗ | ✓ | ✓ |
| RBLA (Tavallaie & Nazemi[1]) | ✓ | ✓ | ✓ | ✓ |
| SmartFed (Wu et al., 2025a) | ✓ | ✗ | ✓ | ✓ |

bias and initialization drift across clients. **FLASC** (Kuo et al., 2024) incorporates sparsity into LoRA to further reduce communication overhead. **SA-FedLoRA** (Yang et al., 2024c) mitigates client drift through parameter regularization and dynamically allocates communication budgets.

**SLoRA** (Babakniya et al., 2023) proposes a novel data-driven initialization scheme to better handle statistical heterogeneity. **RoLoRA** (Chen et al., 2024d) adopts an alternating minimization approach to improve robustness under non-IID conditions. **FedPipe** (Fang et al., 2024b) automatically selects critical parameters for fine-tuning and applies quantization to reduce memory usage. **LP-FL** (Jiang et al., 2023) applies LoRA directly to enable efficient on-device fine-tuning. **Fed-piLot** (Zhang et al., 2024l) reduces memory consumption through LoRA assignment strategies and introduces a novel spatial-temporal aggregation (STAgg) rule to address heterogeneity. **FedRA** (Su et al., 2023) adaptively determines parameter update scopes based on client resource constraints, effectively reducing computational, communication, and memory costs. **FedPruner** (Wu et al., 2025c) introduces a macro-micro synergetic pruning framework to mitigate memory constraints on participating devices. To further facilitate large-scale deployment, **DevFT** (Wu et al., 2025b) proposes a developmental federated tuning paradigm to minimize training resource overheads.

• **Heterogeneous LoRA** allows clients to adopt different rank values $r$ based on their data characteristics or resource constraints. This heterogeneity can manifest either across clients (inter-model) or within different layers of the same model (intra-model). By enabling each client to select a rank that best fits its capabilities and local data, this approach introduces greater flexibility and resource-awareness into the federated fine-tuning process. Table 3 summarizes representative methods and the specific challenges they address.

**HETLoRA** (Cho et al., 2024) assigns heterogeneous ranks across devices and incorporates rank self-pruning along with sparsity-weighted aggregation to tackle data heterogeneity. **FLoRA** (Wang et al., 2024f) proposes a stacking-based aggregation scheme and allows devices to select ranks according to their resource budgets. **FlexLoRA** (Bai et al., 2024a) enables dynamic adjustment of local LoRA ranks to leverage the heterogeneous device resources, while employing singular value decomposition for weight redistribution.

Table 4: **Personalized LoRA in federated fine-tuning.**

| Method | Challenge | | | |
|---|---|---|---|---|
| | Communication | Non-IID | Memory | Computation |
| FDLoRA (Qi et al., 2024a) | ✗ | ✓ | ✗ | ✗ |
| pFedLoRA (Yi et al., 2023) | ✗ | ✓ | ✗ | ✗ |
| FedLoRA (Wu et al., 2024f) | ✗ | ✓ | ✗ | ✗ |
| FedDPA (Yang et al., 2024b) | ✗ | ✓ | ✗ | ✗ |
| PerFIT (Zhang et al., 2024e) | ✗ | ✓ | ✗ | ✗ |
| FedMEM (Du et al., 2024) | ✗ | ✓ | ✗ | ✗ |
| FedAMoLE (Zhang et al., 2024i) | ✗ | ✓ | ✗ | ✗ |

**LoRA-A$^2$** (Koo et al., 2024) introduces alternating freezing and adaptive rank selection mechanisms to fully utilize heterogeneous device resources while addressing statistical heterogeneity. Byun & Lee (2024) propose a replication-based padding technique to enable aggregation across clients with varying LoRA ranks. **FEDHM** (Yao et al., 2021) addresses resource constraints by distributing low-rank models with heterogeneous capacities to clients. **RBLA** (Tavallaie & Nazemi[1]) improves aggregation robustness by simultaneously maintaining and aligning both low-rank and high-rank feature components. **SmartFed** (Wu et al., 2025a) achieves efficient task adaptation by leveraging rank-wise reconfiguration of existing LoRA modules, thereby avoiding the prohibitive computational overhead of training from scratch.

● **Personalized LoRA** enables each participant to fine-tune its model using personalized low-rank adaptation matrices, allowing for better alignment with local data characteristics. This approach enhances the ability of the global model to generalize across clients while retaining client-specific nuances. Table 4 summarizes representative methods and the specific challenges they aim to address.

**FDLoRA** (Qi et al., 2024a) introduces dual LoRA modules on each client to separately capture global and personalized knowledge. **pFedLoRA** (Yi et al., 2023) designs a homogeneous small adapter to facilitate federated clients' heterogeneous local model training, with a proposed iterative training process for global-local knowledge exchange. **FedLoRA** (Wu et al., 2024f) maintains shared general knowledge in a global full-rank matrix while encoding client-specific knowledge in a personalized low-rank module. **FedDPA** (Yang et al., 2024b) utilizes a global adapter and a local adapter to jointly address test-time distribution shifts and client-specific personalization. **PerFIT** (Zhang et al., 2024e) allows each client to search for a personalized architecture by expanding the trainable parameter space of the global model to address data heterogeneity. **FEDMEM** (Du et al., 2024) equips the global model with a KNN classifier that captures client-specific distributional shifts, achieving personalization and overcoming data heterogeneity. **FedAMoLE** (Zhang et al., 2024i) features a mixture of LoRA experts module for aggregating heterogeneous models and a reverse selection-based expert assignment strategy that optimizes model architectures based on data distributions.

## 4.2 Prompt-based Tuning

### 4.2.1 Preliminary

Prompt-based tuning (Lester et al., 2021) has emerged as a highly effective and resource-efficient alternative to conventional fine-tuning approaches for LLMs. Unlike traditional methods that update the model's parameters directly, prompt-based tuning learns a set of trainable prompts or input embeddings that steer the model's behavior on downstream tasks. By modifying only the input space, this approach leverages the pre-trained knowledge of LLMs without altering their weights. As illustrated in Figure 11, a sequence of trainable prompt embeddings $P \in \mathbb{R}^{l_p \times d}$ is prepended to the original input tokens $X \in \mathbb{R}^{l_x \times d}$, where $l_p$ and $l_x$ denote the lengths of the prompt and input sequences, respectively, and $d$ represents the model's hidden dimension. The concatenated sequence is then fed into the frozen model: $Z = f([P; X]; \theta)$, where $f(\cdot; \theta)$ denotes the pre-trained LLM with frozen parameters $\theta$, and $[P; X]$ represents the

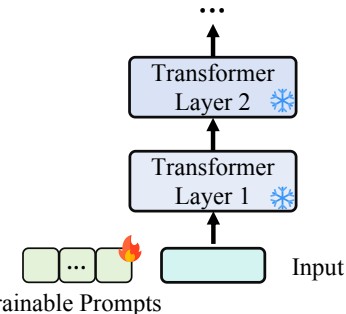

Figure 11: The working principle of prompt tuning.

Table 5: **General prompt tuning in federated fine-tuning.**

| Method | Challenge | | | |
|---|---|---|---|---|
| | Communication | Non-IID | Memory | Computation |
| MetePFL (Chen et al., 2023a) | ✗ | ✗ | ✗ | ✗ |
| PromptFL (Guo et al., 2023c) | ✗ | ✗ | ✗ | ✗ |
| FedBPT (Sun et al., 2023) | ✗ | ✗ | ✓ | ✓ |
| FedPepTAO (Che et al., 2023) | ✓ | ✓ | ✗ | ✗ |
| Fed-BBPT (Lin et al., 2023) | ✗ | ✗ | ✓ | ✓ |
| FedTPG (Qiu et al., 2023) | ✗ | ✗ | ✗ | ✗ |
| FedPR (Feng et al., 2023a) | ✗ | ✓ | ✗ | ✗ |
| Fed-CPrompt (Bagwe et al., 2023) | ✗ | ✓ | ✗ | ✗ |
| Fedprompt (Zhao et al., 2023a) | ✗ | ✗ | ✗ | ✗ |
| FedSP (Dong et al., 2023a) | ✗ | ✗ | ✓ | ✓ |
| HePCo (Halbe et al., 2023) | ✗ | ✓ | ✗ | ✗ |
| PFL-GCN (Ahmad et al., 2023) | ✗ | ✗ | ✗ | ✗ |
| AUG-FedPrompt (Cai et al., 2023b) | ✗ | ✗ | ✗ | ✗ |
| Liu et al. (2023e) | ✗ | ✗ | ✗ | ✗ |
| FedHPL (Ma et al., 2024b) | ✗ | ✓ | ✗ | ✗ |
| PFPT (Weng et al.) | ✗ | ✓ | ✗ | ✗ |
| FCILPT (Liu et al., 2023c) | ✗ | ✓ | ✗ | ✗ |
| CaFPT (Guo et al., 2024c) | ✗ | ✗ | ✗ | ✗ |
| FedPoD (Chen et al., 2023b) | ✗ | ✓ | ✗ | ✗ |

concatenation of the trainable prompts and the original input tokens. By optimizing the prompt embeddings $P$, the model can effectively adapt to new tasks while reusing its pre-trained knowledge. This approach enables efficient task adaptation without modifying the model weights, thereby significantly reducing memory and computational overhead. Through prompt-based tuning, task-specific guidance is embedded within the prompts, allowing the model to generate desired outputs by attending to relevant information stored in its pre-trained parameters.

### 4.2.2 Prompt in Federated Fine-Tuning

In the context of federated fine-tuning, prompt-based tuning offers significant advantages in communication efficiency and model adaptability. Since only the trainable prompt embeddings are updated and exchanged, rather than model parameters, this approach substantially reduces communication overhead between clients and the central server. Additionally, by freezing the base model, prompt-based tuning allows clients with heterogeneous data distributions to personalize their behavior effectively, while still benefiting from globally shared knowledge encoded in the pre-trained model. These properties make prompt-based tuning a compelling choice for federated fine-tuning. In this paper, we propose a novel taxonomy of prompt-based federated fine-tuning approaches, categorizing them into three primary types: General Prompt Tuning, Personalized Prompt Tuning, and Multi-Domain Prompt Tuning, as illustrated in Figure 9. In the following sections, we examine representative methods within each category and analyze how they address the challenges outlined in Section 3, beyond the inherent benefits brought by prompt tuning itself.

• **General Prompt Tuning** refers to approaches in which a shared set of prompt embeddings is learned and applied uniformly across all participating clients. In this setting, the same prompts are prepended to each client's input sequences, providing consistent task-specific guidance and enabling the global model to generalize across diverse data sources. Table 5 summarizes representative methods and the specific challenges they aim to address.

**MetePFL** (Chen et al., 2023a) applies prompt tuning to fine-tune a spatio-temporal Transformer-based foundation model for weather forecasting tasks in a federated setting. **PromptFL** (Guo et al., 2023c) adapts CLIP models for vision-language tasks in FL using prompt-based tuning. **FedBPT** (Sun et al., 2023) employs prompt-based tuning to efficiently adapt black-box LLMs using gradient-free optimization, eliminating the need for clients to access model parameters and requiring only forward propagation for local training. **FedPepTAO** (Che et al., 2023) introduces a partial prompt tuning mechanism to reduce communication costs, along with an adaptive optimization algorithm to address data heterogeneity. **Fed-**

**BBPT** (Lin et al., 2023) enables clients to utilize a zeroth-order optimizer locally, obviating the need for full LLM deployment, effectively reducing memory consumption and computational costs. **FedTPG** (Qiu et al., 2023) learns a unified, task-aware prompt generation network conditioned on input text, improving generalization to both seen and unseen classes.

**FedPR** (Feng et al., 2023a) enhances federated visual prompt tuning by projecting local prompt updates into an approximate null space of the global prompt, mitigating gradient interference and improving global performance. **Fed-CPrompt** (Bagwe et al., 2023) addresses asynchronous task arrivals and heterogeneous data distributions via asynchronous prompt updates and a contrastive continual learning loss. **FedPrompt** (Zhao et al., 2023a) employs a split aggregation strategy, freezing the extensive parameters of LLMs and only tuning and aggregating soft prompts. **FedSP** (Dong et al., 2023a) reduces computational and memory overhead by utilizing a lightweight auxiliary model for prompt learning. **HePCo** (Halbe et al., 2023) mitigates catastrophic forgetting and data heterogeneity through a data-free distillation method performed in the model's latent space. **PFL-GCN** (Ahmad et al., 2023) employs prompt tuning specifically for sentiment analysis.

**AUG-FedPrompt** (Cai et al., 2023b) exploits abundant unlabeled data for data augmentation to address the issue of data scarcity. Liu et al. (2023e) integrate self-consistency and chain-of-thought prompting to improve zero-shot performance of LLMs. **FedHPL** (Ma et al., 2024b) introduces a global logit distillation framework to handle model heterogeneity and guide the local training process. **PFPT** (Weng et al.) proposes a probabilistic prompt aggregation mechanism to address data heterogeneity and imbalanced data distribution. **FCILPT** (Liu et al., 2023c) jointly encodes task-relevant and task-irrelevant knowledge into prompts to preserve both previous and newly learned knowledge, alleviating catastrophic forgetting. **CaFPT** (Guo et al., 2024c) leverages information-theoretic principles to facilitates the retrieval process by conditioning on examples that activate the most relevant knowledge inside pre-trained models. **FedPoD** (Chen et al., 2023b) employs lightweight prompts to guide frozen foundation models and introduces multi-level prompt-based communication to enable multi-source knowledge fusion and controlled optimization.

• **Personalized Prompt Tuning** enables each client to tailor its prompt embeddings based on local data distributions and task-specific requirements. By fine-tuning prompts individually, clients can better capture local nuances and context-specific information that a one-size-fits-all prompt might overlook. This approach directly addresses the challenge of data heterogeneity by facilitating local adaptation, while still allowing clients to benefit from global knowledge aggregated during training. Table 6 summarizes representative methods and the specific challenges they target.

**pFedPG** (Yang et al., 2023a) deploys a personalized prompt generator on the server to produce client-specific visual prompts, enabling efficient adaptation of frozen backbones to diverse local data. **SGPT** (Deng et al., 2024) combines generalized and personalized FL by learning a mix of shared and group-specific prompts to capture both commonalities and group-specific variations. **pFedPrompt** (Guo et al., 2023b) leverages the unique multimodal capabilities of vision-language models by learning client consensus in the linguistic space and adapting to client characteristics in the visual space in a non-parametric manner. **FedOTP** (Li et al., 2024c) introduces efficient collaborative prompt learning strategies to capture diverse category traits on a per-client basis. **pFedPT** (Li et al., 2023b) utilizes personalized visual prompts to implicitly represent local data distribution information and provides this information to the aggregation model to enhance classification tasks. **FedMGP** (Yu et al., 2024) uses coarse-grained global prompts for shared knowledge and fine-grained local prompts for personalization, and introduces a selective fusion mechanism for prompt aggregation.

**FedLPPA** (Lin et al., 2024) jointly learns personalized prompts and aggregation strategies for weakly-supervised medical image segmentation. **FedPGP** (Cui et al., 2024) employs pre-trained CLIP to provide knowledge-guidance for the global prompt, enhancing generalization while incorporating a low-rank adaptation term to personalize the global prompt. **FedPFT** (Wu et al., 2024g) addresses feature-classifier mismatch through prompt-driven feature transformation. Wang et al. (2024c) propose a discrete local search strategy for gradient-free local training and a token-based compression method inspired by linear word analogies, substantially reducing resource costs. **pFedMoAP** (Luo et al., 2024) introduces a personalized prompt learning framework based on the mixture-of-experts paradigm (Cai et al., 2024a). **CP$^2$GFed** (Gao et al., 2024a) introduces a cross-granularity knowledge transfer mechanism and dynamic personalized prompt generation to improve model performance.

Table 6: **Personalized prompt tuning in federated fine-tuning.**

| Method | Challenge | | | |
|---|---|---|---|---|
| | Communication | Non-IID | Memory | Computation |
| pFedPG (Yang et al., 2023a) | ✗ | ✓ | ✗ | ✗ |
| SGPT (Deng et al., 2024) | ✗ | ✓ | ✗ | ✗ |
| pFedPrompt (Guo et al., 2023b) | ✗ | ✓ | ✗ | ✗ |
| FedOTP (Li et al., 2024c) | ✗ | ✓ | ✗ | ✗ |
| pFedPT (Li et al., 2023b) | ✗ | ✓ | ✗ | ✗ |
| FedMGP (Yu et al., 2024) | ✗ | ✓ | ✗ | ✗ |
| FedLPPA (Lin et al., 2024) | ✗ | ✓ | ✗ | ✗ |
| FedPGP (Cui et al., 2024) | ✗ | ✓ | ✗ | ✗ |
| FedPFT (Wu et al., 2024g) | ✗ | ✓ | ✗ | ✗ |
| Wang et al. (2024c) | ✓ | ✓ | ✓ | ✓ |
| pFedMoAP (Luo et al., 2024) | ✗ | ✓ | ✗ | ✗ |
| CP$^2$GFed (Gao et al., 2024a) | ✗ | ✓ | ✗ | ✓ |

Table 7: **Multi-domain prompt tuning in federated fine-tuning.**

| Method | Challenge | | | |
|---|---|---|---|---|
| | Communication | Non-IID | Memory | Computation |
| DiPrompT (Bai et al., 2024b) | ✗ | ✓ | ✗ | ✗ |
| PFCR (Guo et al., 2024a) | ✗ | ✓ | ✗ | ✗ |
| Fed-DPT (Wei et al., 2023) | ✗ | ✓ | ✗ | ✗ |
| FedAPT (Su et al., 2024) | ✗ | ✓ | ✗ | ✗ |
| Zhao et al. (2024b) | ✗ | ✓ | ✗ | ✗ |
| FedDG (Gong et al., 2024) | ✗ | ✓ | ✗ | ✗ |
| CP-Prompt (Feng et al., 2024b) | ✗ | ✓ | ✗ | ✗ |

• **Multi-Domain Prompt Tuning** extends the prompt-based approach to environments where federated clients operate across distinct domains or application contexts. In such scenarios, each client is equipped with domain-specific prompt embeddings that adapt the shared global model to diverse contextual and distributional conditions. This approach enhances the model's generalization ability across heterogeneous domains while maintaining a shared global foundation. It is particularly valuable in real-world deployments spanning multiple industries or task categories. Table 7 summarizes representative methods and the specific challenges they address.

**DiPrompT** (Bai et al., 2024b) proposes a distributed domain generalization approach using adaptive prompts, introducing global prompts for shared knowledge and domain prompts for domain-specific adaptation. **PFCR** (Guo et al., 2024a) eliminates the need for raw data sharing via encrypted gradient updates, models items in a unified feature space using descriptive text, and facilitates cross-domain knowledge transfer through federated content representations and prompt tuning. **Fed-DPT** (Wei et al., 2023) leverages a pretrained vision-language model and applies dual prompt tuning—combining visual and textual prompts—for improved domain alignment across decentralized data sources. **FedAPT** (Su et al., 2024) introduces a meta prompt, an adaptive network, and frozen keys to personalize prompts for each test sample, thereby enhancing multi-domain image classification. Zhao et al. (2024b) propose a language distance metric to improve data efficiency and facilitate cross-linguistic generalization. **FedDG** (Gong et al., 2024) allows clients to learn text and visual prompts locally while maintaining indirect alignment via global prompts used as a shared reference. Domain-specific prompts are exchanged among clients and selectively integrated into global prompts using lightweight attention-based aggregators. **CP-Prompt** (Feng et al., 2024b) captures intra-domain knowledge by inserting personalized prompts into the multi-head attention modules and subsequently learns inter-domain representations through a shared prompting mechanism.

## 4.3 Adapter-based Tuning

### 4.3.1 Preliminary

Adapter-based tuning is another parameter-efficient alternative to full-parameter fine-tuning for LLMs (Pfeiffer et al., 2020). It introduces lightweight, trainable adapter modules into the model while keeping the pre-trained weights frozen. These modules act as task-specific components that transform intermediate representations in a controlled manner, enabling efficient adaptation to downstream tasks with minimal memory and computational overhead. A standard adapter module consists of three key operations: **down-projection**, **non-linearity**, and **up-projection**, as shown in Figure 12. For activations $h_i \in \mathbb{R}^{n \times d}$, the adapter transformation proceeds as follows:

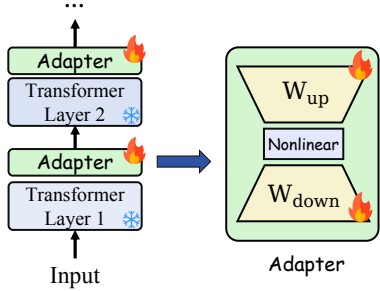

Figure 12: The working principle of adapter tuning.

**1) Down-Projection:** The high-dimensional hidden state $h_i$ is projected into a low-dimensional space using a learnable weight matrix $W_{DP} \in \mathbb{R}^{d \times r}$, where $r \ll d$ controls the bottleneck size. This step reduces the number of trainable parameters while capturing essential features:

$$h_i^{'} = h_i W_{DP} \tag{4}$$

**2) Non-Linearity:** A non-linear activation function $\sigma(\cdot)$, such as ReLU or GELU, is applied to introduce expressive transformations while retaining important task-specific patterns:

$$h_i^{''} = \sigma(h_i^{'}) \tag{5}$$

**3) Up-Projection:** The transformed low-dimensional representation is mapped back to the original feature space using an **up-projection** matrix $W_{UP} \in \mathbb{R}^{r \times d}$:

$$h_i^{'''} = h_i^{''} W_{UP} \tag{6}$$

**4) Residual Connection:** The final output of the adapter is then residually added to the original hidden state, preserving the pre-trained knowledge while incorporating task-specific adjustments:

$$Z = h_i + h_i^{'''} \tag{7}$$

where $Z$ represents the adapted hidden representation. This residual connection ensures that the pre-trained model remains largely intact while allowing task-specific fine-tuning through the lightweight adapter layers. Notably, only the adapter parameters $W_{DP}$ and $W_{UP}$ are updated during training, resulting in significantly lower memory and computation costs compared to full-parameter fine-tuning.

### 4.3.2 Adapter in Federated Fine-Tuning

In the context of federated fine-tuning, adapter-based tuning provides significant advantages in both resource efficiency and model adaptability. Since only the lightweight adapter modules are updated and exchanged, rather than the full model parameters, this approach greatly reduces computation and communication overhead. Moreover, by freezing the base model, adapter-based tuning enables clients to fine-tune efficiently on heterogeneous local data while still benefiting from the shared global knowledge encoded in the pre-trained model. This modular design facilitates the seamless integration of task-specific adaptations without compromising the generalization capability of the base model. To better understand the landscape of adapter-based methods in federated fine-tuning, we propose a new taxonomy comprising three categories: General Adapter Tuning, Personalized Adapter Tuning, and Multi-Domain Adapter Tuning, as illustrated in Figure 9. In the following sections, we explore representative methods within each category and analyze how they address the core challenges identified in Section 3, beyond the inherent benefits brought by adapter itself.

● **General Adapter Tuning** refers to scenarios in which all clients utilize a shared adapter structure with identical initialization. In this setting, the same adapter modules are inserted into the transformer layers of each client's model, enabling consistent adaptation mechanisms across the federation. This uniformity facilitates stable aggregation and coordinated updates during federated training. Such an approach is particularly effective when clients operate on similar tasks or share relatively homogeneous data distributions,

Table 8: **General, personalized, and multi-domain adapter tuning in federated fine-tuning.**

| Method | Type | Challenge | | | |
|---|---|---|---|---|---|
| | | Communication | Non-IID | Memory | Computation |
| FedAdapter (Cai et al., 2023a) | General | ✓ | ✗ | ✗ | ✓ |
| Kim et al. (2023a) | General | ✓ | ✓ | ✓ | ✓ |
| FedTT+ (Ghiasvand et al., 2024) | General | ✓ | ✓ | ✗ | ✗ |
| C2A (Kim et al., 2023b) | Personalized | ✗ | ✓ | ✗ | ✗ |
| FedCLIP (Lu et al., 2023b) | Personalized | ✗ | ✓ | ✗ | ✗ |
| Fed-MNMT (Liu et al., 2023f) | Multi-domain | ✗ | ✓ | ✗ | ✗ |
| AdaFedSelecKD (Feng et al., 2024a) | Multi-domain | ✗ | ✓ | ✗ | ✗ |
| FedDAT (Chen et al., 2024a) | Multi-domain | ✗ | ✓ | ✗ | ✗ |

as a globally optimized adapter can generalize well across participants. Table 8 summarizes representative methods and the specific challenges they aim to address.

**FedAdapter** (Cai et al., 2023a) proposes a progressive adapter tuning strategy, combined with continuous device profiling, to dynamically optimize adapter configurations across clients, improving efficiency without sacrificing accuracy. Kim et al. (2023a) leverage adapters to address the high communication costs associated with federated fine-tuning of LLMs. **FedTT+** (Ghiasvand et al., 2024) integrates tensorized adapters for LLM adaptation and further improves robustness to data heterogeneity by freezing portions of the tensor factors, significantly reducing the number of trainable parameters while maintaining model performance.

• **Personalized Adapter Tuning** enables each client to independently fine-tune its adapter modules based on its local data distribution and task-specific requirements. In contrast to general adapter tuning, this approach does not enforce uniformity across clients; instead, it allows for the retention of personalized adapter parameters that better capture client-specific knowledge. This strategy is particularly advantageous in federated settings characterized by high degrees of data heterogeneity. By leveraging personalized adapters, clients can achieve improved local performance while still benefiting from shared global knowledge. Table 8 summarizes representative methods and the specific challenges they address.

**C2A** (Kim et al., 2023b) employs a hypernetwork to generate client-specific adapters, effectively addressing data heterogeneity by enabling on-demand parameter generation tailored to each client. **FedCLIP** (Lu et al., 2023b) introduces an attention-based adapter design that utilizes the pre-trained model's knowledge to facilitate both rapid generalization and efficient personalization while minimizing resource overhead.

• **Multi-Domain Adapter Tuning** extends the federated fine-tuning paradigm to clients operating across distinct domains, enabling efficient adaptation to domain-specific tasks. In this setting, each client maintains its own domain-specific adapter while contributing to a shared global model. The global model aggregates adapter updates across domains to capture domain-invariant representations to support generalization. This approach is particularly effective in cross-domain scenarios such as multilingual natural language processing. By decoupling domain-specific learning from the shared backbone, this strategy balances personalization and collaboration. Table 8 summarizes representative methods and the challenges they address.

**Fed-MNMT** (Liu et al., 2023f) applies adapter-based fine-tuning for multilingual neural machine translation, significantly reducing communication overhead. It further explores parameter clustering strategies to mitigate conflicts during aggregation. **AdaFedSelecKD** (Feng et al., 2024a) performs adapter-based summarization to minimize transmitted parameters and introduces selective knowledge distillation for efficient domain-adaptive learning. **FedDAT** (Chen et al., 2024a) proposes a dual-adapter teacher framework to regularize local updates under data heterogeneity, and employs mutual knowledge distillation for effective cross-client knowledge transfer.

### 4.4 Selective-based Tuning

#### 4.4.1 Preliminary

Selective-based tuning has emerged as an efficient strategy for fine-tuning LLMs by updating specific parameters of the model while keeping the majority of pre-trained weights frozen (Kornblith et al., 2019). This

approach significantly reduces computational and memory overhead compared to full-parameter fine-tuning, while maintaining the model's generalization ability. Among selective-based tuning techniques, two widely adopted strategies are bias tuning (Zaken et al., 2021) and partial tuning (Houlsby et al., 2019), both of which optimize only a subset of parameters rather than the entire model.

Bias tuning updates only the bias terms of the model while keeping all other parameters frozen. Despite its simplicity, this approach has demonstrated strong performance across a variety of tasks with minimal overhead. Partial tuning generalizes this idea by allowing updates to a carefully selected subset of model parameters, such as layer normalization parameters, feed-forward network biases, or specific attention blocks. By focusing updates on the most relevant parameters, selective-based tuning methods improve training efficiency, mitigate catastrophic forgetting, and enable rapid adaptation using limited data and resources.

### 4.4.2 Selective Fine-Tuning Methods

**DP-BiTFiT** (Bu et al., 2022) applies differentially private bias-term tuning in centralized training scenarios to ensure privacy-preserving adaptation while reducing resource demands. **FedPEFT** (Sun et al., 2022) shares only a small subset of model weights, such as bias parameters, significantly reducing the communication overhead. **RaFFM** (Yu et al., 2023d) introduces a resource-aware model compression framework tailored for FL, which includes salient parameter prioritization and subnetwork extraction to support dynamic model scaling across heterogeneous edge devices. Sun et al. (2024) propose a selective layer-wise fine-tuning approach to reduce training cost while preserving model performance.

## 4.5 Other Tuning Methods

In addition to mainstream fine-tuning strategies, several alternative approaches have been explored to optimize LLMs in FL. These methods primarily include **zeroth-order optimization** (Chen et al., 2017), **split learning** (Thapa et al., 2022), **model compression** (Choudhary et al., 2020), and **data selection** (Shen, 2024) (as shown in Figure 9). For example, **FedKSeed** (Qin et al., 2023b) employs zeroth-order optimization with a finite set of random seeds, enabling LLM fine-tuning without storing intermediate activations and reducing communication overhead. **FedBERT** (Tian et al., 2022b) combines FL with split learning to pre-train BERT in a distributed, privacy-preserving manner, achieving efficient model training across decentralized clients. **FedBiOT** (Wu et al., 2024b) compresses the LLM at the server side while allowing clients to fine-tune lightweight adapters, significantly reducing resource consumption. **FedHDS** (Qin et al., 2024) introduces a hierarchical data selection framework that identifies representative coresets for instruction tuning, minimizing redundancy at both intra- and inter-client levels to improve training efficiency in FL.

# 5 Datasets and Benchmarks

A comprehensive evaluation framework is essential for systematically assessing the effectiveness and generalization ability of federated fine-tuning methods. In this section, we begin by presenting widely-used fine-tuning datasets spanning multiple domains. We then detail a suite of domain-specific evaluation benchmarks that enable consistent, fine-grained, and standardized assessment of FedLLM performance across diverse and heterogeneous task settings.

## 5.1 Instruction Fine-Tuning Datasets

Table 9 and Table 10 present a comprehensive collection of instruction fine-tuning datasets spanning several key domains, including general language understanding, finance, medicine, code, math, and law. For each domain, we select representative datasets from Hugging Face that are widely adopted by the community and closely aligned with real-world FedLLM applications. Building on this overview, we next provide a detailed introduction to the datasets within each domain.

### 5.1.1 General Instruction Fine-Tuning Datasets

General instruction fine-tuning datasets are primarily designed to improve the overall instruction-following capability of LLMs across a wide range of tasks and domains (Zhou et al., 2023). These datasets serve as a foundation for aligning LLMs with human intent, and are particularly useful in federated settings where clients may engage in diverse yet general-purpose interactions.

For English instruction fine-tuning, representative datasets include Alpaca (Taori et al., 2023), which is generated by Text-Davinci-003 with Alpaca-style instruction prompts, and Alpaca-GPT4 (Peng et al., 2023), which builds on this with GPT-4-generated, multi-turn dialogues to improve linguistic nuance and contextual coherence. Other commonly used datasets include Self-Instruct (Wang et al., 2022b), UltraChat 200k (Ding et al., 2023a), OpenOrca (Lian et al., 2023), ShareGPT-90K[1], WizardLM Evol-Instruct V2 196k (Xu et al., 2023a), Databricks Dolly 15K (Conover et al., 2023), Baize (Xu et al., 2023b), OpenChat (Wang et al., 2023a), and Flan-v2[2].

For Chinese instruction tuning, the landscape includes representative datasets such as BELLE-CN (Ji et al., 2023a), which focuses on improving model alignment with Chinese user instructions. This is complemented by additional resources like Firefly-train-1.1M (Yang, 2023), Wizard-LM-Chinese-Instruct-Evol (Xu et al., 2023a), and HC3-Chinese (Guo et al., 2023a). These resources facilitate a diverse range of instruction tasks, including reasoning, question answering, and domain-specific knowledge modeling.

Bilingual instruction datasets are also included to support multilingual and cross-lingual instruction tuning in federated contexts. HC3 (Guo et al., 2023a) offers both human- and model-generated responses in English and Chinese for evaluating factual consistency and detection capabilities. ShareGPT-Chinese-English-90k (shareAI, 2023) provides high-quality, bilingual conversations, making it particularly suitable for instruction tuning in multilingual FedLLM scenarios. A detailed comparison is provided in Table 9.

### 5.1.2 Financial Domain Instruction Fine-Tuning Datasets

Instruction fine-tuning datasets in the financial domain are tailored to equip language models with specialized knowledge and reasoning capabilities relevant to financial tasks (Li et al., 2023h). In the context of FedLLM, such datasets are particularly valuable for enabling privacy-preserving and institution-specific applications, including investment recommendation, market sentiment analysis, financial reporting assistance, and regulatory compliance. These use cases often involve sensitive and proprietary data that cannot be centrally aggregated due to privacy, confidentiality, or regulatory constraints (Byrd & Polychroniadou, 2020), making federated fine-tuning an ideal solution.

For English instruction tuning, representative datasets include FinGPT (Zhang et al., 2023a), which provides a large corpus of financial question-answering pairs and document summaries tailored to real-world financial analysis. Finance-Instruct-500k (Flowers, 2025) and Finance-Alpaca[3] extend general instruction-tuning formats to financial scenarios, offering instruction–response pairs related to stock prediction, portfolio analysis, and macroeconomic commentary. Additional datasets such as Financial PhraseBank (Malo et al., 2014), Yahoo-Finance-Data[4], Financial-QA-10K[5], Financial-Classification[6], Twitter-Financial-News-Topic[7], Financial-News-Articles[8], and FiQA (Maia et al., 2018) cover a broad spectrum of financial tasks including sentiment classification, time-series event extraction, and financial question answering. Transcripts from earnings calls further support fine-tuning for multi-turn, dialogue-based financial reasoning.

For Chinese financial applications, Doc2EDAG (Zheng et al., 2019) provides a rich dataset for event detection and argument generation from financial documents, supporting instruction-style tasks. In addition, the Synthetic-PII-Finance-Multilingual dataset (Watson et al., 2024) includes multi-language synthetic financial

---

[1]https://huggingface.co/datasets/liyucheng/ShareGPT90K
[2]https://huggingface.co/datasets/SirNeural/flan__v2
[3]https://huggingface.co/datasets/gbharti/finance-alpaca
[4]https://huggingface.co/datasets/bwzheng2010/yahoo-finance-data
[5]https://huggingface.co/datasets/virattt/financial-qa-10K
[6]https://huggingface.co/datasets/nickmuchi/financial-classification
[7]https://huggingface.co/datasets/zeroshot/twitter-financial-news-topic
[8]https://huggingface.co/datasets/ashraq/financial-news-articles

Table 9: A summary of representative instruction fine-tuning datasets in **general, financial, and medical** domains. The construction methods are categorized into three types: human construct, which refers to datasets written or annotated by humans; model construct, which refers to data generated by prompting LLMs; and synthetic, which refers to data produced through rule-based or programmatic generation.

| Dataset | Language | Construction | Domain | Description |
|---|---|---|---|---|
| Alpaca | English | Model | General | Generated by Text-Davinci-003 (Alpaca-style) |
| Alpaca-GPT4 | English | Model | General | Generated by GPT-4 (multi-turn Alpaca prompts) |
| Self-Instruct | English | Human + Model | General | Generated from seed instructions via GPT-3 |
| UltraChat 200k | English | Model | General | Multi-turn dialogues filtered from UltraChat |
| OpenOrca | English | Model | General | 4.2M GPT-3.5/4 augmented FLAN examples |
| ShareGPT90K | English | Model | General | 90K multi-turn dialogues from ShareGPT |
| WizardLM Evol-Instruct V2 | English | Model | General | 196K samples generated via Evol-Instruct |
| Databricks Dolly 15K | English | Human | General | 15K human-generated prompt-response pairs |
| Baize | English | Model | General | Dialogues from ChatGPT (prompted by user queries) |
| OpenChat | English | Model | General | Dialogues from open LLMs for multi-turn alignment |
| Flan-v2 | English | Model | General | Combination of Flan, P3, SNI, CoT, and Dialog tasks |
| BELLE-CN | Chinese | Human + Model | General | 2M Chinese instruction-following samples |
| Firefly-train-1.1M | Chinese | Human | General | 1.65M Chinese samples (23 tasks, human templates) |
| Wizard-LM-Chinese-instruct-evol | Chinese | Human + Model | General | 70K Chinese samples (translated from WizardLM) |
| HC3-Chinese | Chinese | Human + Model | General | Chinese Human-ChatGPT QA pairs (multi-domain) |
| HC3 | English / Chinese | Human + Model | General | Human-ChatGPT QA pairs (multi-domain) |
| ShareGPT-Chinese-English-90k | English / Chinese | Model | General | 90K bilingual human-machine QA pairs |
| FinGPT | English | Human + Model | Financial | Instruction-tuning data for financial tasks |
| Finance-Instruct-500k | English | Human + Model | Financial | Large-scale instruction-tuning data for financial tasks |
| Finance-Alpaca | English | Human + Model | Financial | Instruction-tuning data combining Alpaca and FiQA |
| Financial PhraseBank | English | Human | Financial | Manually annotated financial news for sentiment |
| Yahoo-Finance-Data | English | Human | Financial | Yahoo finance data |
| Financial-QA-10K | English | Model | Financial | Contextual QA for financial answering and retrieval |
| Financial-Classification | English | Human | Financial | Financial PhraseBank and Kaggle financial texts |
| Twitter-Financial-News-Topic | English | Human | Financial | Annotated tweets for financial topic classification |
| Financial-News-Articles | English | Human | Financial | Financial articles for classification and sentiment analysis |
| FiQA | English | Human | Financial | QA from financial texts and forums |
| Earnings-Call | English | Human | Financial | QA pairs from CEO/CFO earnings calls |
| Doc2EDAG | Chinese | Human | Financial | Financial reports annotated for event graph extraction |
| Synthetic-PII-Finance-Multilingual | Multilingual | Synthetic | Financial | Synthetic financial docs with labeled PII |
| ChatDoctor-200K | English | Human + Model | Medical | Medical QA and dialogue instructions |
| ChatDoctor-HealthCareMagic-100k | English | Human | Medical | Real-world doctor-patient conversations |
| Medical Meadow CORD-19 | English | Human | Medical | Literature summarization instructions (CORD-19) |
| Medical Meadow MedQA | English | Human | Medical | Medical QA from professional board exams |
| HealthCareMagic-100k-en | English | Human | Medical | Medical and real doctor-patient consultations |
| ChatMed-Consult-Dataset | Chinese | Human + Model | Medical | Medical consultation QA and dialogue instructions |
| CMtMedQA | Chinese | Human | Medical | Multi-turn real medical QA |
| DISC-Med-SFT | Chinese | Human + Model | Medical | Real dialogues and knowledge graph QA pairs |
| HuatuoGPT-SFT | Chinese | Human + Model | Medical | ChatGPT-generated and real doctor-patient dialogues |
| Huatuo26M-Lite | Chinese | Human + Model | Medical | Refined subset of Huatuo-26M (ChatGPT-rewritten) |
| ShenNong-TCM | Chinese | Human + Model | Medical | QA pairs from TCM knowledge graph |
| MedDialog | English / Chinese | Human | Medical | Large-scale doctor-patient dialogues |

records annotated with privacy-related attributes. A comparative summary of these finance-related datasets is presented in Table 9.

### 5.1.3 Medical Domain Instruction Fine-Tuning Datasets

Instruction fine-tuning datasets in the medical domain are designed to enable LLMs to perform tasks such as medical reasoning, patient interaction, and clinical decision support (Zhang et al., 2023h). Within the context of FedLLM, these datasets are particularly relevant for privacy-preserving healthcare applications, where sensitive patient data is inherently distributed across hospitals, clinics, and personal health devices, and cannot be centrally aggregated due to strict privacy regulations (Nguyen et al., 2022). Federated fine-tuning with medical instruction data empowers models to generalize across diverse clinical intents while preserving the confidentiality and heterogeneity of local medical records, thereby supporting robust and compliant deployment in real-world healthcare settings.

For English-language datasets, ChatDoctor-200K (Li et al., 2023i) and ChatDoctor-HealthCareMagic-100k (Li et al., 2023i) contain medical dialogues derived from professional consultation platforms, facilitating multi-turn reasoning and symptom analysis. The Medical Meadow (Han et al., 2023) suite offers curated

Table 10: A summary of representative instruction fine-tuning datasets in **code, math, and law** domains.

| Dataset | Language | Construction | Domain | Description |
|---------|----------|--------------|--------|-------------|
| CodeAlpaca | English | Model | Code | GPT-generated code instructions (Alpaca-style) |
| Code Instructions 120k Alpaca | English | Human + Model | Code | Code generation instructions (Alpaca-style) |
| CodeContests | English | Human | Code | Competitive programming problems |
| CommitPackFT | English | Human | Code | Filtered GitHub commits with high-quality messages |
| ToolBench | English | Human + Model | Code | Instructions for multi-tool API usage |
| CodeParrot | English | Human | Code | Deduplicated and filtered Python files from GitHub |
| The Stack v2 Dedup | English | Human | Code | Large-scale deduplicated source code |
| CodeSearchNet | English | Human | Code | Code-language pairs for retrieval and semantic search |
| CodeForces-CoTs | English | Human + Model | Code | 10k CodeForces problems with CoT (DeepSeek R1) |
| CodeXGLUE Code Refinement | English | Human | Code | Buggy and fixed Java functions for code refinement |
| GSM8K | English | Human | Math | 8.5k grade school math word problems |
| CoT-GSM8k | English | Human + Model | Math | GSM8K extended with CoT reasoning steps |
| MathInstruct | English | Human + Model | Math | CoT and Program-of-Thought rationales |
| MetaMathQA | English | Human + Model | Math | Question augmentations from GSM8K and MATH |
| OpenR1-Math-220k | English | Human + Model | Math | 220k math problems with CoT (DeepSeek R1) |
| Competition MATH | English | Human | Math | 12.5K high school competition math problems |
| DeepMind Mathematics Dataset | English | Synthetic | Math | Algorithmically generated math problems |
| OpenMathInstruct-1 | English | Human + Model | Math | 1.8M math problems with code-interpreter solutions |
| Orca-Math Word Problems 200k | English | Synthetic | Math | 200K synthetic grade school math word problems |
| DAPO-Math-17k | English | Human + Model | Math | 17K diverse math problems for reasoning |
| Big-Math-RL-Verified | English | Human + Model | Math | 251K math problems with verifiable answers for RL |
| BELLE-math-zh | Chinese | Human + Model | Math | Chinese elementary school math problems |
| MathInstruct-Chinese | Chinese | Model | Math | Chinese version of MathInstruct |
| Legal-QA-v1 | English | Human | Law | 3.7K legal QA pairs from legal forums |
| Pile of Law | English | Human | Law | Dataset for legal-domain tasks |
| CUAD | English | Human | Law | 26K expert-annotated legal contract QA |
| LEDGAR | English | Human | Law | 1.45M expert-labeled legal contract clauses |
| DISC-Law-SFT | Chinese | Human + Model | Law | Comprehensive instructions for diverse legal tasks |
| Law-GPT-zh | Chinese | Human | Law | Legal sentence pairs for embedding models |
| Lawyer LLaMA | Chinese | Human + Model | Law | Dataset for Chinese legal tasks |

datasets from sources such as CORD-19[9] and MedQA[10], targeting tasks like biomedical question answering, literature summarization, and clinical fact verification. Additionally, HealthCareMagic-100k-en[11] supports English patient–doctor interactions across various medical specialties.

For Chinese medical instruction tuning, a rich set of datasets has been developed to address the unique linguistic and clinical characteristics of Chinese healthcare scenarios. Notable examples include ChatMed-Consult-Dataset (Zhu et al., 2023b) and CMtMedQA (Yang et al., 2024a), which focus on consultation-style QA pairs. DISC-Med-SFT (Bao et al., 2023), HuatuoGPT-SFT (Li et al., 2023d), and Huatuo26M-Lite[12] offer large-scale instruction–response pairs in clinical medicine, public health, and disease treatment. Traditional Chinese Medicine (TCM) is also covered through datasets like ShenNong-TCM (Zhu & Wang, 2023), enabling LLMs to support specific diagnostic and treatment tasks.

The MedDialog (Zeng et al., 2020) dataset provides bilingual (English and Chinese) medical dialogues, supporting cross-lingual instruction tuning and evaluation in federated medical environments where linguistic diversity and clinical protocol variance are prevalent. A comparative summary of these medical-specific datasets, including language, construction method, and description, is provided in Table 9.

### 5.1.4 Code Domain Instruction Fine-Tuning Datasets

Instruction fine-tuning datasets in the code domain are curated to improve a language model's ability to understand, generate, and reason about source code across various programming languages and tasks (Muennighoff et al., 2023). In the context of FedLLM, such datasets are particularly valuable for enabling privacy-preserving applications like on-device programming assistants, secure code generation within enterprise environments, and personalized developer support. Given that source code repositories often contain proprietary algorithms, sensitive business logic, or embedded credentials, federated fine-tuning offers a compelling al-

---

[9]https://huggingface.co/datasets/medalpaca/medical_meadow_cord19
[10]https://huggingface.co/datasets/medalpaca/medical_meadow_medqa
[11]https://huggingface.co/datasets/wangrongsheng/HealthCareMagic-100k-en
[12]https://huggingface.co/datasets/FreedomIntelligence/Huatuo26M-Lite

ternative to centralized training on raw code, allowing organizations to harness LLM capabilities without compromising code confidentiality.

Representative datasets include CodeAlpaca (Chaudhary, 2023) and Code Instructions 120k Alpaca[13], which extend the Alpaca instruction format to software engineering tasks such as debugging, function generation, and refactoring. CodeContests (Li et al., 2022c) focuses on competitive programming tasks and provides instruction–response pairs related to algorithmic problem solving. CommitPackFT (Muennighoff et al., 2023) offers commit-message generation tasks based on source code diffs, reflecting real-world software maintenance scenarios. ToolBench (Qin et al., 2023a) is designed to help models learn tool-augmented code generation through instruction-following examples.

Several large-scale pretraining and fine-tuning datasets are also widely used for code instruction alignment. CodeParrot[14] and The Stack v2 Dedup (Kocetkov et al., 2022) provide diverse and deduplicated code corpora across multiple programming languages, while CodeSearchNet (Husain et al., 2019) and CodeXGLUE (Lu et al., 2021) support retrieval, summarization, and translation tasks with instruction-style prompts. CodeForces-CoTs (Penedo et al., 2025) incorporates chain-of-thought annotations for programming tasks, supporting more explainable and step-wise code generation. These code-specific datasets play a vital role in developing FedLLMs that can support privacy-sensitive, domain-specific programming environments. A comparative summary of these code-related datasets is presented in Table 10.

### 5.1.5 Math Domain Instruction Fine-Tuning Datasets

Instruction fine-tuning datasets in the math domain are developed to enhance a language model's proficiency in mathematical reasoning, symbolic computation, and step-by-step problem solving (Tang et al., 2024b). These datasets are particularly valuable in FedLLM scenarios such as personalized math tutoring, intelligent educational platforms, and localized STEM applications, where sensitive student information or institution-specific curricular content must remain on-device. Fine-tuning LLMs in the math domain poses unique challenges, as it requires not only a strong grasp of language but also precise logical inference and numerical accuracy—skills essential for generating correct and interpretable mathematical solutions.

For English-language datasets, GSM8K (Cobbe et al., 2021) is a widely used benchmark for grade-school math word problems with detailed rationales. CoT-GSM8K[15] augments it with chain-of-thought explanations to support intermediate reasoning steps. Datasets such as MathInstruct (Yue et al., 2023b), MetaMathQA (Yu et al., 2023b), OpenR1-Math-220k (Allal et al., 2025), and Orca-Math Word Problems 200k (Mitra et al., 2024) provide high-quality, instruction-based math problems covering arithmetic, algebra, and word problem solving. The Competition MATH Benchmark (Hendrycks et al., 2021c) and DeepMind Mathematics[16] Dataset offer more advanced, competition-style problems suitable for evaluating formal mathematical reasoning. Recent datasets like DAPO-Math-17k (Yu et al., 2025), Big-Math-RL-Verified (Albalak et al., 2025), and OpenMathInstruct-1 (Toshniwal et al., 2024) integrate reinforcement signals, verifiable proofs, or multi-step derivations, further pushing the boundaries of instruction-aligned mathematical LLMs.

For Chinese-language instruction tuning datasets, BELLE-math-zh[17] and MathInstruct-Chinese[18] provide diverse mathematical problems adapted to Chinese curricula and linguistic structures. These datasets facilitate the training and evaluation of FedLLMs in multilingual educational contexts, supporting privacy-preserving and culturally contextualized math assistance. A comparative summary of these math-focused instruction tuning datasets is provided in Table 10.

### 5.1.6 Legal Domain Instruction Fine-Tuning Datasets

Instruction fine-tuning datasets in the legal domain are designed to improve a language model's capability in legal reasoning, contract analysis, statute interpretation, and other tasks that require deep, domain-specific

---

[13]https://huggingface.co/datasets/iamtarun/code_instructions_120k_alpaca
[14]https://huggingface.co/datasets/codeparrot/codeparrot-clean
[15]https://huggingface.co/datasets/Dahoas/cot_gsm8k
[16]https://huggingface.co/datasets/di-zhang-fdu/DeepMind_Mathematics_QA
[17]https://huggingface.co/datasets/frankminors123/belle-math-zh
[18]https://huggingface.co/datasets/ALmonster/MathInstruct-Chinese

understanding of legal language and structure (Yue et al., 2023a). In the context of FedLLM, these datasets are particularly important for enabling decentralized legal assistance systems, confidential contract review, and on-device compliance monitoring—applications where legal data is often sensitive, jurisdiction-bound, and subject to strict confidentiality constraints. As centralized training on legal documents is frequently infeasible due to regulatory and privacy concerns, federated fine-tuning offers a promising approach to leveraging LLMs in legal settings without compromising data security or legal integrity.

For English-language instruction tuning, Legal-QA-v1[19] provides question–answer pairs covering legal concepts and procedures across various subfields of law. Pile of Law (Gao et al., 2020) is a large-scale corpus of U.S. legal documents—including court opinions, contracts, and regulations—that supports open-ended legal instruction tuning. CUAD (Hendrycks et al., 2021b) focuses on contract understanding, with annotated question–answer pairs tailored for clause extraction and risk analysis. LEDGAR[20] contains a large set of contractual clauses categorized into fine-grained legal functions, useful for classification and retrieval tasks in instruction-based formats.

For Chinese-language legal modeling, DISC-Law-SFT (Yue et al., 2023a) offers supervised instruction–response pairs across a wide spectrum of Chinese legal domains, including civil law, criminal law, and administrative law. Law-GPT-zh[21] consolidates multiple sources of Chinese legal texts into an instruction tuning format to support legal consultation and statutory reasoning. Lawyer LLaMA (Huang et al., 2023a) further augments legal dialogue capabilities with simulated lawyer–client conversations, making it well-suited for on-device legal assistants in federated environments. A comparative overview of these law-specific instruction tuning datasets is provided in Table 10.

## 5.2 Evaluation Benchmarks

### 5.2.1 General Evaluation Benchmarks

General-purpose evaluation benchmarks play a critical role in systematically assessing the instruction-following ability, reasoning competence, and overall robustness of LLMs across a wide range of tasks and domains (Lou et al., 2024). In the context of FedLLM, such benchmarks are especially valuable for evaluating model generalization under heterogeneous data distributions, identifying robustness gaps in decentralized training settings, and facilitating consistent comparisons between personalized and globally aggregated models. Existing benchmarks can be broadly categorized into four groups: (i) general reasoning and instruction-following, (ii) robustness, alignment, and meta-evaluation, (iii) multilingual and Chinese-specific benchmarks, and (iv) long-context understanding. These benchmarks provide a foundation for rigorous and reproducible evaluation of FedLLM across heterogeneous and dynamic environments.

**(i) General reasoning and instruction-following.** This category includes evaluation benchmarks such as MMLU (Hendrycks et al., 2020), BIG-bench (Srivastava et al., 2022), DROP (Dua et al., 2019), CRASS (Frohberg & Binder, 2021), and ARC (Clark et al., 2018), which assess multitask knowledge, discrete reasoning, and science question answering. AGIEval (Zhong et al., 2023), M3Exam (Zhang et al., 2023f), and SCIBENCH (Wang et al., 2023d) extend evaluation to standardized exams and college-level math, physics, and chemistry. Instruction-following quality is addressed by Vicuna Evaluation[22], MT-Bench (Zheng et al., 2023a), AlpacaEval (Dubois et al., 2023), Chatbot Arena (Zheng et al., 2023a), and PandaLM (Wang et al., 2023f). Datasets like HellaSwag (Zellers et al., 2019) and TruthfulQA (Lin et al., 2021) focus on commonsense inference and factual accuracy, respectively.

Several benchmarks target more specialized reasoning abilities: ScienceQA (Lu et al., 2022a) evaluates multimodal scientific question answering; Chain-of-Thought Hub (Fu et al., 2023) focuses on step-wise reasoning; NeuLR (Xu et al., 2023c) benchmarks deductive, inductive, and abductive reasoning; ALCUNA (Yin et al., 2023) assesses generalization to novel knowledge; LMExamQA (Bai et al., 2023c) tests models on recall, understanding, and analysis across over academic questions. Benchmarks like SocKET (Choi et al., 2023) and Choice-75 (Hou et al., 2023) address social knowledge and decision-making in scripted scenarios, respectively.

---

[19]https://huggingface.co/datasets/dzunggg/legal-qa-v1
[20]https://huggingface.co/datasets/coastalchp/ledgar
[21]https://huggingface.co/datasets/sentence-transformers/law-gpt
[22]https://github.com/lm-sys/vicuna-blog-eval

Table 11: Overview of **general** evaluation benchmarks. Abbreviations: EM (Exact Match), OOD (Out-of-Distribution), NER (Named Entity Recognition), ICL (In-Context Learning), RE (Relation Extraction), MRR (Mean Reciprocal Rank), ASR (Adversarial Success Rate).

| Benchmark | Domain | Evaluation Objective | Main Evaluation Criteria |
|---|---|---|---|
| MMLU | Gen. | Multitask understanding (57 subjects) | Accuracy |
| BIG-bench | Gen. | Advanced reasoning capabilities | EM, ROUGE, and Accuracy |
| DROP | Gen. | Discrete reasoning over paragraphs | EM & F1 |
| CRASS | Gen. | Counterfactual reasoning | Accuracy |
| ARC | Gen. | Grade-school science reasoning | Accuracy |
| AGIEval | Gen. | Performance on human-centric exams | Accuracy |
| M3Exam | Gen. | Multilingual, multimodal, multilevel reasoning | Accuracy |
| SCIBENCH | Gen. | College-level scientific problem-solving | Accuracy & Error Attribution |
| Vicuna Evaluation | Gen. | Instruction-following quality | Human/GPT-4 Preference |
| MT-Bench | Gen. | Multi-turn conversation & instruction-following | GPT-4 Win Rate |
| AlpacaEval | Gen. | Instruction-following (LLM auto-annotation) | Win Rate |
| Chatbot Arena | Gen. | Human-preference | Elo Score |
| PandaLM | Gen. | Instruction-following & hyperparameter impact | PandaLM Win Rate |
| HellaSwag | Gen. | Commonsense inference (continuation selection) | Accuracy |
| TruthfulQA | Gen. | Truthfulness & falsehood avoidance | Truthfulness Rate & Accuracy |
| ScienceQA | Gen. | Multimodal scientific reasoning & explanation | Accuracy & Explanation Quality |
| CoT Hub | Gen. | Multi-step reasoning via CoT | Accuracy |
| NeuLR | Gen. | Deductive, inductive, and abductive reasoning | Accuracy |
| ALCUNA | Gen. | Novel knowledge comprehension & reasoning | Accuracy |
| LMExamQA | Gen. | Knowledge recall, understanding, and analysis | Accuracy |
| SocKET | Gen. | Social knowledge understanding | Accuracy |
| Choice-75 | Gen. | Decision reasoning (scripted scenarios) | Accuracy |
| HELM | Gen. | Holistic evaluation (multi-metric scenarios) | Composite Score |
| OpenLLM | Gen. | Open-style reasoning (multi-benchmark) | Normalized Accuracy |
| BOSS | Gen. | OOD robustness | OOD Accuracy Drop |
| GLUE-X | Gen. | OOD robustness | OOD Accuracy Drop |
| PromptBench | Gen. | Robustness & prompt-engineering | ASR & Accuracy |
| DynaBench | Gen. | Robustness (dynamic human-in-loop) | Error Rate |
| KoLA | Gen. | Evolving world knowledge (19 tasks) | Self-Contrast Calibration |
| CELLO | Gen. | Complex instruction-following | Accuracy & BLEU |
| LLMEval | Gen. | Meta-evaluation of LLM evaluators | Meta-evaluator Agreement |
| Xiezhi | Gen. | Holistic domain knowledge (516 disciplines) | MRR |
| C-Eval | Gen. | Chinese domain knowledge & reasoning | Accuracy |
| BELLE-eval | Gen. | Chinese instruction-following & multi-skill | GPT-4 Win Rate |
| SuperCLUE | Gen. | Chinese instruction-following (human-aligned) | GPT-4 Win Rate & Accuracy |
| M3KE | Gen. | Chinese knowledge (71 disciplines, 4 levels) | Accuracy |
| BayLing-80 | Gen. | Cross-lingual & conversational capabilities | GPT-4 Win Rate |
| MMCU | Gen. | Multitask Chinese understanding | Accuracy |
| C-CLUE | Gen. | Classical Chinese NER & RE | Accuracy, Recall, and F1 |

Broad-scoped platforms such as HELM (Liang et al., 2022) and OpenLLM[23] integrate multiple datasets and offer normalized aggregate scores across diverse metrics and tasks.

**(ii) Robustness, alignment, and meta-evaluation.** To simulate the variability and noise of federated environments, benchmarks such as BOSS (Yuan et al., 2023), GLUE-X (Yang et al., 2022), PromptBench (Zhu et al., 2023a), and DynaBench (Kiela et al., 2021) test model robustness to input distribution shifts, prompt perturbations, and adversarial examples. KoLA (Yu et al., 2023a) emphasizes world knowledge calibration, while CELLO (He et al., 2024) further tests compliance under real-world constraints. LLMEval (Zhang et al., 2023g) functions as a meta-evaluation benchmark to assess the consistency and reliability of LLM-based evaluators, an important consideration when deploying automated evaluation in federated settings.

**(iii) Multilingual and Chinese-specific benchmarks.** Benchmarks such as Xiezhi (Gu et al., 2024b), C-Eval (Huang et al., 2023b), BELLE-eval (Ji et al., 2023b), SuperCLUE (Xu et al., 2023d), M3KE (Liu et al., 2023a), BayLing-80 (Zhang et al., 2023d), MMCU (Zeng, 2023), and C-CLUE[24] provide a diverse

---

[23]https://huggingface.co/spaces/open-llm-leaderboard/open_llm_leaderboard
[24]https://github.com/jizijing/C-CLUE

Table 12: Overview of **general long-context (LC)** evaluation benchmarks.

| Benchmark | Domain | Evaluation Objective | Main Evaluation Criteria |
|---|---|---|---|
| LongBench | Gen. (LC) | Bilingual long-context understanding | Accuracy, F1, and ROUGE |
| L-Eval | Gen. (LC) | Long-context reasoning (up to 60k) | Accuracy & Win Rate |
| InfiniteBench | Gen. (LC) | Long-context understanding (up to 2M) | Accuracy & ROUGE |
| Marathon | Gen. (LC) | Long-context understanding (multi-domain) | Accuracy |
| LongEval | Gen. (LC) | Long-context retrieval | Accuracy |
| BABILong | Gen. (LC) | Long-context reasoning (haystack) | Accuracy |
| DetectiveQA | Gen. (LC) | Long-context reasoning | Accuracy & GPT-4 Judge Score |
| NoCha | Gen. (LC) | Narrative comprehension | Accuracy |
| Loong | Gen. (LC) | Document reasoning & extended-context QA | GPT-4 Judge Score |
| TCELongBench | Gen. (LC) | Temporal reasoning over long narratives | Accuracy |
| DENIAHL | Gen. (LC) | Context feature influence | ROUGE & EM |
| LongMemEval | Gen. (LC) | Long-term interactive memory | EM |
| Long2RAG | Gen. (LC) | Long-text RAG capabilities | BLEU, ROUGE, and EM |
| L-CiteEval | Gen. (LC) | Citation & evidence utilization | ROUGE & Accuracy |
| LIFBENCH | Gen. (LC) | Long-context instruction-following | Accuracy, ROUGE, and Pass@k |
| LongReason | Gen. (LC) | Long-context reasoning | Accuracy & ROUGE-L |
| BAMBOO | Gen. (LC) | Long-text modeling (diverse tasks) | Accuracy, F1, and ROUGE |
| ETHIC | Gen. (LC) | Long-context understanding | F1, Win Rate, and EM |
| LooGLE | Gen. (LC) | Long-context understanding and reasoning | GPT-4 Judge Score & Auto Metric |
| HELMET | Gen. (LC) | Multi-skill long-context capabilities | Task Metrics & Human Evaluation |
| HoloBench | Gen. (LC) | Holistic reasoning (database-style text) | Accuracy |
| LOFT | Gen. (LC) | Long-context reasoning | Accuracy, EM, and Recall |
| Lv-Eval | Gen. (LC) | Long-context comprehension (up to 256k) | Keyword Recall & F1 |
| ManyICLBench | Gen. (LC) | Many-shot ICL capabilities | Accuracy |
| ZeroSCROLLS | Gen. (LC) | Zero-shot inference capabilities | ROUGE & F1 |
| LongICLBench | Gen. (LC) | Long-context ICL | Accuracy, ROUGE-L, and Pass@k |
| LIBRA | Gen. (LC) | Russian long-context understanding | Accuracy, Recall, and F1 |

evaluation landscape for Chinese and multilingual LLMs. These benchmarks span topics ranging from academic disciplines to sociocultural reasoning, using metrics including GPT-4 preference scoring, multitask accuracy, F1 scores, and normalized Elo rankings.

**(iv) Long-context understanding.** Long-context reasoning is critical for FedLLM applications involving document-intensive tasks, extended dialogue, and memory retention. Benchmarks such as LongBench (Bai et al., 2023b), L-Eval (An et al., 2023), InfiniteBench (Zhang et al., 2024h), Marathon (Zhang et al., 2023b), LongEval (Li et al., 2023a), and BABILong (Kuratov et al., 2024) form the backbone of this category, measuring comprehension, retrieval, and reasoning over contexts up to 2 million tokens. Further specialized benchmarks include: 1) Narrative and temporal reasoning: DetectiveQA (Xu et al., 2024e), NoCha (Karpinska et al., 2024), Loong (Wang et al., 2024b) and TCELongBench (Zhang et al., 2024k) focus on long-range narrative understanding and event-based temporal reasoning in complex textual sequences. 2) Retrieval and memory evaluation: DENIAHL (Dai et al., 2024), LongMemEval (Wu et al., 2024a), Long2RAG (Qi et al., 2024b), and L-CiteEval (Tang et al., 2024a) assess in-context feature sensitivity, long-term memory utilization, retrieval grounding, and the model's ability to incorporate external citations.

3) Instruction-following and generation under long input: LIFBENCH (Wu et al., 2024e), LongReason (Ling et al., 2025), BAMBOO (Dong et al., 2023b), ETHIC (Lee et al., 2024b), and LooGLE (Li et al., 2023e) evaluate multi-criteria instruction-following accuracy, response stability, and coherence under extended prompts. HELMET (Yen et al., 2024) further integrates summarization, retrieval, and reasoning in unified evaluation pipelines, supporting both automatic and human metrics. 4) Database-style and structured input reasoning: HoloBench (Maekawa et al., 2024) and LOFT (Lee et al., 2024a) benchmark the ability to perform complex reasoning over structured or database-like inputs, including table querying, execution accuracy, and factual consistency, especially in scenarios where retrieval-augmented methods are substituted by long-context modeling.

5) Scaling and generalization with longer context: Lv-Eval (Yuan et al., 2024b), ManyICLBench (Zou et al., 2024), ZeroSCROLLS (Shaham et al., 2023), and LongICLBench (Li et al., 2024f) examine the scalability of in-context learning and multi-task alignment as input length increases, making them valuable tools for

analyzing the performance ceiling of long-context FedLLMs. 6) Cross-lingual long-context evaluation: LI-BRA (Churin et al., 2024) tests instruction-following and long-form coherence in Russian, contributing to the evaluation of multilingual long-context capabilities. These long-context benchmarks are crucial for validating the scalability and persistence of FedLLMs, especially in environments requiring private document analysis, extended user sessions, or continual knowledge tracking. Table 11 and Table 12 summarize the general evaluation benchmarks discussed above.

### 5.2.2 Financial Domain Evaluation Benchmarks

Finance-specific evaluation benchmarks are crucial for assessing the domain alignment, factual accuracy, and reasoning capabilities of LLMs in high-stakes financial contexts (Li et al., 2023h). In federated settings, where sensitive data from banks, asset managers, and regulatory bodies cannot be centralized due to confidentiality and compliance constraints, these benchmarks serve as vital tools for evaluating model performance under decentralized, privacy-preserving, and task-diverse conditions. Effective financial evaluation must encompass a broad range of tasks—including open-book question answering, multi-step quantitative reasoning, and document classification—while also accounting for linguistic nuances, regulatory requirements, and the structural complexity of financial texts. Such benchmarks are indispensable for ensuring the robustness and reliability of FedLLM in real-world financial applications.

**(i) Multi-task and agent-style financial evaluation.** Several benchmarks offer broad-spectrum evaluation across multiple financial tasks, aligning well with FedLLM's need to support diverse client use cases (e.g., auditing, compliance, trading). FinBen (Xie et al., 2024) evaluates holistic financial reasoning over 24 tasks using automatic metrics, retrieval-augmented generation accuracy, and expert human judgment. PIXIU (Xie et al., 2023), FLUE (Shah et al., 2022), and BBT-CFLEB (Lu et al., 2023a) benchmark LLMs on multi-task setups including sentiment classification, QA, event detection, and stock prediction. CFinBench (Nie et al., 2024) and SuperCLUEFin (Xu et al., 2024a) extend this paradigm to Chinese financial scenarios, assessing models across regulatory knowledge, certification preparation, and real-world task instructions using multi-type question formats. ICE-PIXIU (Xie et al., 2023) further supports bilingual Chinese–English evaluation, suitable for cross-regional FedLLM deployments. FLARE-ES (Zhang et al., 2024g) enables bilingual Spanish–English testing to evaluate cross-lingual transfer and domain-specific reasoning.

**(ii) Task-specific capability evaluation.** Fine-grained benchmarks evaluate specific financial NLP capabilities that are critical for real-world FedLLM deployment, including document question answering, information extraction, and numerical reasoning. FinanceBench (Islam et al., 2023) targets open-book question answering using real-world, company-related financial documents, with a focus on factual correctness and evidence alignment—a critical requirement for FedLLM deployed in compliance, auditing, and enterprise document analysis scenarios. FiNER-ORD (Shah et al., 2023) and FinRED (Sharma et al., 2022) evaluate financial Named Entity Recognition and relation extraction from news and earnings transcripts—key for localized data processing on devices with limited connectivity. FinQA (Chen et al., 2021b) and BizBench (Koncel-Kedziorski et al., 2023) benchmark multi-step numerical and quantitative reasoning, involving both tabular data and executable financial logic, useful in portfolio analysis, valuation, and budgeting applications. Econ-LogicQA (Quan & Liu, 2024) addresses sequential economic reasoning over multi-event scenarios. In the Chinese financial domain, FinEval (Zhang et al., 2023c) and CFBenchmark (Lei et al., 2023) assess task-specific instruction following across topics such as taxation, accounting, and investment strategy. Hirano (Hirano, 2024) enables financial language understanding evaluation in Japanese, while MultiFin (Jørgensen et al., 2023) focuses on multilingual topic classification, useful in federated settings with cross-border clients.

**(iii) Long-context financial reasoning.** Many practical financial tasks involve extended documents such as annual reports, investor briefs, and regulatory filings. Long-context benchmarks are critical to evaluate whether FedLLM can perform document-level comprehension and multi-step derivation under memory and privacy constraints. DocFinQA (Reddy et al., 2024) simulates multi-step numerical reasoning over financial reports, measuring exact match, F1, and reasoning traceability. FinTextQA (Chen et al., 2024b) further tests open-ended QA over long textual contexts using BLEU and ROUGE for generation quality. These benchmarks reflect realistic federated scenarios, such as on-device due diligence support or local regulatory interpretation, where global models must adapt to long-form content without accessing raw documents. A

Table 13: Overview of evaluation benchmarks in the **financial** and **medical** domains.

| Benchmark | Domain | Evaluation Objective | Main Evaluation Criteria |
|---|---|---|---|
| FinBen | Financial | Holistic financial capabilities | Auto metrics & Human eval |
| PIXIU | Financial | Financial NLP tasks | Accuracy & F1 |
| FLUE | Financial | Financial language understanding | Accuracy, F1, nDCG, and MRR |
| BBT-CFLEB | Financial | Chinese financial understanding and generation | ROUGE, F1, and Accuracy |
| CFinBench | Financial | Chinese financial knowledge | Accuracy |
| SuperCLUEFin | Financial | Chinese financial assistant capabilities | Win Rate & Accuracy |
| ICE-PIXIU | Financial | Bilingual (Chinese–English) financial reasoning | Accuracy, F1, and ROUGE |
| FLARE-ES | Financial | Bilingual (Spanish–English) financial reasoning | Accuracy, F1, and ROUGE |
| FinanceBench | Financial | Financial open-book QA (real-world) | Factual Correctness |
| FiNER-ORD | Financial | Financial NER | F1, Precision, and Recall |
| FinRED | Financial | Financial RE (news, transcripts) | F1, Precision, and Recall |
| FinQA | Financial | Numerical reasoning over financial reports | Accuracy |
| BizBench | Financial | Quantitative reasoning on realistic problems | Accuracy & F1 |
| EconLogicQA | Financial | Economic sequential reasoning | Accuracy |
| FinEval | Financial | Chinese financial knowledge and reasoning | Accuracy |
| CFBenchmark | Financial | Chinese financial assistant capabilities | LLM Judge Score & Accuracy |
| Hirano | Financial | Japanese financial understanding | Accuracy |
| MultiFin | Financial | Multilingual financial topic classification | F1 & Accuracy |
| DocFinQA | Financial (LC) | Long-context financial document reasoning | Accuracy |
| FinTextQA | Financial (LC) | Long-form financial QA | ROUGE & LLM Judge Score |
| CBLUE | Medical | Chinese biomedical understanding | Accuracy & F1 |
| PromptCBLUE | Medical | Chinese medical knowledge (prompt-based) | Accuracy & F1 |
| CMB | Medical | Chinese medical knowledge | Accuracy & LLM Judge Score |
| HuaTuo26M | Medical | Chinese medical knowledge and QA | ROUGE & and LLM Judge Score |
| CMExam | Medical | Chinese medical licensing exam | Accuracy, BLEU, and ROUGE |
| MultiMedQA | Medical | Clinical knowledge and open-ended QA | Accuracy & Human Evaluation |
| QiZhenGPT | Medical | Drug indication identification | Accuracy |
| MedExQA | Medical | Medical knowledge and explanation generation | Accuracy & LLM judge Score |
| JAMA and Medbullets | Medical | Challenging clinical QA | Accuracy & Human Evaluation |
| MedXpertQA | Medical | Expert-level medical reasoning (multi-modal) | Accuracy |
| MedJourney | Medical | Full clinical patient journey tasks | Accuracy, ROUGE, BLEU, and F1 |
| MedAgentsBench | Medical | Complex multi-step clinical reasoning | Accuracy |
| LongHealth | Medical (LC) | QA over long-form clinical documents | Accuracy |
| MedOdyssey | Medical (LC) | Long-context medical understanding | Accuracy, Recall, and F1 |

summary of these finance-specific evaluation benchmarks, including their evaluation objectives and corresponding metrics, is presented in Table 13.

### 5.2.3 Medical Domain Evaluation Benchmarks

Medical-specific evaluation benchmarks are essential for assessing the performance of LLMs in healthcare applications, where accuracy, safety, and domain-specific understanding are critical (Thirunavukarasu et al., 2023). In the context of FedLLM, these benchmarks are particularly important for evaluating models deployed in privacy-sensitive environments, such as hospital intranets, personal health monitoring devices, and clinical support systems, where patient data cannot be centralized due to regulatory and ethical constraints. To reflect the complexity of medical reasoning and language comprehension under such federated conditions, medical benchmarks cover a wide range of evaluation targets, including diagnostic reasoning, medical question answering, clinical document understanding, and guideline adherence.

**(i) Medical knowledge and QA evaluation.** This category focuses on assessing LLMs' general and specialized medical knowledge through structured question-answering tasks. CBLUE (Zhang et al., 2021) and PromptCBLUE[25] benchmark Chinese biomedical NLP tasks across information extraction, document classification, and QA, with the latter emphasizing prompt-based generation. CMB (Wang et al., 2023e), HuaTuo26M (Li et al., 2023d), and CMExam (Liu et al., 2023d) evaluate models on Chinese medical exam-style QA, clinical diagnosis tasks, and real-world query understanding. These benchmarks are well-suited for evaluating FedLLM tailored to localized clinical documentation and public health information. Multi-MedQA (Singhal et al., 2023) serves as a high-quality English benchmark encompassing multiple-choice and open-ended medical QA, with expert-based evaluation of factuality, helpfulness, and safety—three pillars

---

[25]https://github.com/michael-wzhu/PromptCBLUE

critical for deploying FedLLM in patient-facing applications. QiZhenGPT[26] provides an annotation-based evaluation of drug indication extraction from natural language prompts, useful for drug interaction checking at the edge. MedExQA (Kim et al., 2024) expands QA evaluation to underrepresented medical specialties, while JAMA and Medbullets (Chen et al., 2025) challenge models with high-difficulty US medical exam-style questions and demand strong explanation quality.

**(ii) Clinical reasoning and care process modeling.** In practice, many medical applications require multi-step diagnostic reasoning, care pathway modeling, and treatment planning, especially in multi-agent or longitudinal clinical scenarios. MedXpertQA (Zuo et al., 2025) tests both textual and multimodal reasoning, simulating specialist-level question answering across medical images and textual reports. MedJourney (Wu et al., 2024d) evaluates LLMs across the full patient care pipeline, from chief complaint triage to follow-up guidance, with both task-specific and human expert evaluation. MedAgentsBench (Tang et al., 2025) explicitly focuses on multi-turn clinical planning, assessing the model's ability to generate coherent and correct diagnostic and treatment steps over multiple interactions—an ideal setup for privacy-preserving agent-based FedLLM deployment.

**(iii) Long-context clinical understanding.** Federated medical applications often involve lengthy patient histories, radiology reports, or guideline documents. Benchmarks in this group evaluate the ability of LLMs to perform robust QA and inference over long-form inputs. LongHealth (Adams et al., 2024) focuses on QA over long clinical narratives, measuring exact match and F1 accuracy. MedOdyssey (Fan et al., 2024a) targets long-context understanding across clinical specialties, incorporating ROUGE, EM, and human preference scoring to assess consistency and informativeness over extended reasoning chains. A comprehensive summary of these medical evaluation benchmarks is provided in Table 13.

### 5.2.4 Code Domain Evaluation Benchmarks

Code-specific evaluation benchmarks are essential for assessing the ability of FedLLM to understand, generate, and reason over programming logic across multiple languages and task types (Jiang et al., 2024a). In federated settings, code-related applications are commonly deployed in heterogeneous environments, including personalized coding assistants, on-device tutoring systems, and localized software development workflows. These scenarios place unique demands on LLMs: they must follow fine-grained instructions, ensure functional correctness, and maintain high reliability—all while operating under the resource constraints and privacy requirements that are characteristic of FedLLM deployment. Benchmarks in this domain typically evaluate code synthesis, completion, bug fixing, and multi-turn problem solving.

**(i) Basic code generation and functional reasoning.** This group of benchmarks focuses on evaluating models' ability to generate syntactically correct and semantically valid code from natural language instructions. HumanEval (Chen et al., 2021a) and MBPP (Austin et al., 2021) are foundational benchmarks evaluating one-shot Python function generation, judged via execution-based metrics like pass@k. APPS (Hendrycks et al., 2021a) and DS-1000 (Lai et al., 2023) extend this to more complex and real-world tasks, with DS-1000 focusing on data science scenarios across multiple Python libraries. CodeXGLUE (Lu et al., 2021) offers a broad suite of tasks—spanning code summarization, translation, and generation—making it suitable for assessing FedLLM that may specialize in different sub-tasks across clients. CruxEval (Gu et al., 2024a) further emphasizes logical reasoning and error-free execution in high-stakes contexts. ODEX (Wang et al., 2022c) introduces cross-lingual code generation, evaluating the model's ability to translate natural language into code in four different programming languages, a valuable benchmark for multilingual FedLLM deployment. These datasets are particularly useful for evaluating client-level specialization in federated setups, where users might work with domain-specific codebases or tools.

**(ii) Multi-turn synthesis and structural understanding.** Modern code development involves iterative and structural logic, making multi-turn and component-aware benchmarks crucial. MTPB (Nijkamp et al., 2022) focuses on multi-turn program synthesis, assessing the ability to generate partial, compositional sub-programs in sequence. ClassEval (Du et al., 2023) evaluates whether LLMs can generate coherent Python classes, including method dependencies and variable interactions—reflecting real-world object-oriented programming needs. BigCodeBench (Zhuo et al., 2024) challenges models with complex, multi-functional in-

---

[26]https://github.com/CMKRG/QiZhenGPT

struction following, and rich code behavior evaluation, using criteria like test case accuracy and branch coverage. HumanEvalPack (Muennighoff et al., 2023) extends HumanEval's principles to six languages and multiple code-related subtasks, enabling multilingual federated evaluation. BIRD (Li et al., 2023f) adds a structured dimension by testing text-to-SQL generation for database querying, emphasizing schema-aware reasoning and executable correctness—an increasingly relevant task in enterprise AI agents and personal data querying under private data constraints. Such benchmarks are particularly relevant for FedLLM deployed in collaborative or enterprise environments, where partial programs must be incrementally refined by agents or users with limited compute.

**(iii) Long-context code understanding and retrieval.** Real-world software development often requires reasoning over extended codebases or repositories, posing a challenge for memory-constrained clients. RepoQA (Liu et al., 2024b) and LongCodeArena (Bogomolov et al., 2024) evaluate models' comprehension and retrieval-augmented reasoning over entire repositories or large code documents. They measure not only token-level accuracy but also structural coherence and retrieval efficacy. These benchmarks offer valuable insights into the performance of long-context-aware FedLLM on realistic software engineering tasks such as bug fixing, documentation generation, and legacy code comprehension—particularly under constraints imposed by limited local computational resources.

Together, these code-specific benchmarks provide a comprehensive foundation for assessing the capabilities and limitations of FedLLM in coding applications. A detailed overview of benchmark objectives and evaluation metrics is provided in Table 14.

### 5.2.5 Math Domain Evaluation Benchmarks

Mathematical reasoning tasks serve as a rigorous benchmark for evaluating the generalization, compositionality, and step-by-step problem-solving capabilities of LLMs (Ahn et al., 2024). In federated settings, math-specific evaluations are especially valuable for applications such as privacy-preserving intelligent tutoring systems, on-device educational tools, and personalized STEM learning assistants. These tasks present unique challenges for FedLLM, as they require precise multi-step reasoning, symbolic computation, and logical consistency—often under strict memory, computation, and communication constraints. As such, math benchmarks are instrumental in assessing a model's ability to perform structured reasoning in resource-constrained and heterogeneous environments.

**(i) Primary math reasoning across educational levels.** Benchmarks in this group assess basic-to-advanced math problem solving and reasoning: GSM8K (Cobbe et al., 2021) focuses on grade school arithmetic word problems, serving as a foundation for reasoning evaluation. Competition MATH (Hendrycks et al., 2021c) extends this to competition-level questions in algebra, geometry, and calculus, emphasizing step-by-step derivation accuracy. MathOdyssey (Fang et al., 2024a) evaluates reasoning across high school, university, and Olympiad levels, providing a broad difficulty spectrum. MathBench (Liu et al., 2024a) systematically tests both theoretical understanding and practical application across five levels, reflecting multi-tier federated education scenarios. CHAMP (Mao et al., 2024) introduces concept and hint annotations, useful for federated tutoring agents that may require step-wise guidance or personalization for struggling learners. LILA (Mishra et al., 2022) further expands evaluation to 23 math task types, offering comprehensive insight into the model's versatility across mathematical formats.

**(ii) Formal proof and symbolic reasoning.** Mathematics often requires formal logic and symbolic structure, which tests a model's ability to generalize beyond pattern-matching: MiniF2F (Zheng et al., 2021) and ProofNet (Azerbayev et al., 2023) evaluate formal mathematical reasoning and proof generation, with the latter using Lean 3 as a backend for correctness verification. AlphaGeometry (Trinh et al., 2024) blends neural and symbolic reasoning for Euclidean geometry, a domain requiring precise spatial logic and theorem synthesis—especially relevant in expert-centric or research-level FedLLM deployment.

**(iii) Visual and diagrammatic mathematical reasoning.** Many real-world math problems include visual elements (graphs, tables, geometric diagrams), posing multi-modal reasoning challenges: MathVerse (Zhang et al., 2024f) and We-Math (Qiao et al., 2024) evaluate visual reasoning using diagram interpretation, requiring fine-grained attention to layout and symbolic grounding. U-MATH (Chernyshev et al., 2024) tests open-ended university-level questions involving visual cues, with LLM-assisted expert

Table 14: Overview of evaluation benchmarks in the **code**, **math**, and **law** domains.

| Benchmark | Domain | Evaluation Objective | Main Evaluation Criteria |
|---|---|---|---|
| HumanEval | Code | Code generation and algorithmic reasoning | Pass@k |
| MBPP | Code | Basic Python code generation | Pass@k |
| APPS | Code | Coding challenge competence | Pass@k |
| DS-1000 | Code | Data science code generation | Accuracy & Pass@k |
| CodeXGLUE | Code | Code understanding and generation | Accuracy, BLEU, and MRR |
| CruxEval | Code | Code reasoning, understanding, and execution | Pass@k |
| ODEX | Code | Code generation and execution | Pass@k |
| MTPB | Code | Multi-turn program synthesis | Pass@k |
| ClassEval | Code | Class-level code generation | Pass@k |
| BigCodeBench | Code | Complex and practical code generation | Pass@k |
| HumanEvalPack | Code | Multilingual code tasks | Pass@k |
| BIRD | Code | Database-grounded text-to-SQL | Execution Accuracy & F1 |
| RepoQA | Code (LC) | Long-context code understanding | Pass Rate |
| LongCodeArena | Code (LC) | Long-context code tasks | Pass@k, Accuracy, and EM |
| GSM8K | Math | Grade school math reasoning | Accuracy |
| Competition MATH | Math | Competition math problem-solving | Accuracy |
| MathOdyssey | Math | High-level math problem-solving (Olympiad) | Accuracy |
| MathBench | Math | Comprehensive and hierarchical math evaluation | Accuracy & Circular Evaluation |
| CHAMP | Math | Fine-grained math reasoning | Accuracy |
| LILA | Math | Meta-benchmark for math reasoning | Accuracy & Pass@k |
| MiniF2F | Math | Formal math reasoning (Olympiad) | Proof Accuracy |
| ProofNet | Math | Auto-formalization and formal proof generation | Accuracy & Success Rate |
| AlphaGeometry | Math | Neuro-symbolic reasoning (Euclidean geometry) | Solve Rate |
| MathVerse | Math | Visual math reasoning | Accuracy & CoT Score |
| We-Math | Math | Multi-step math reasoning | Accuracy |
| U-MATH | Math | Open-ended visual math (university-level) | LLM Judge Score |
| TabMWP | Math | Math reasoning over text and tabular data | Accuracy |
| MathHay | Math (LC) | Long-context math reasoning (multi-step) | Accuracy |
| LegalBench | Law | Legal reasoning across six tasks | Accuracy, F1, and ROUGE |
| LexGLUE | Law | Legal language understanding | Accuracy & F1 |
| LEXTREME | Law | Multilingual/multitask legal understanding | Accuracy & F1 |
| LawBench | Law | Chinese legal reasoning | Accuracy & F1 |
| LAiW | Law | Chinese legal tasks | Accuracy & F1 |
| LexEval | Law | Chinese legal understanding and reasoning | Accuracy |
| CitaLaw | Law | Legal answer generation | MRR, Recall, and ROUGE |
| LegalAgentBench | Law | LLM agent legal task-solving | Success Rate & LLM Judge Score |
| SCALE | Law (LC) | Multilingual/long-document legal reasoning | ROUGE & METEOR |

scoring. TabMWP (Lu et al., 2022b) focuses on text–table joint reasoning, simulating practical applications like report analysis or financial tutoring in federated agents.

**(iv) Long-context mathematical reasoning.** FedLLM deployed on real-world devices often faces scenarios where mathematical problems span multiple steps or documents: MathHay (Wang et al., 2024a) evaluates reasoning across extended input chains, testing memory retention and logic consistency in multi-hop math reasoning—an important benchmark for long-context capabilities in private and offline educational settings.

Together, these benchmarks form a robust and diverse suite for evaluating the mathematical competency of FedLLM under different input formats, difficulty levels, and reasoning demands. They are especially vital for personalized STEM learning assistants and edge-based automated math tutoring. A comprehensive summary of these math evaluation benchmarks is provided in Table 14.

### 5.2.6  Legal Domain Evaluation Benchmarks

Legal AI applications impose stringent requirements on factual accuracy, contextual understanding, and logical consistency—making legal benchmarks particularly important for evaluating FedLLM designed for use in areas such as smart justice, personalized legal consultation, and privacy-preserving regulatory compliance. These benchmarks are designed to assess model capabilities across a range of legal reasoning tasks, including legal text comprehension, argument analysis, statutory interpretation, and multi-document synthesis.

Moreover, they often adopt multilingual and long-context formats that reflect the real-world complexity and heterogeneity of legal documents—challenges that are amplified in federated settings (Chen et al., 2024e).

**(i) Foundational and legal-specific reasoning.** Benchmarks in this category assess general-purpose legal understanding, including statute interpretation, legal judgment tasks, and document comprehension: LegalBench (Guha et al., 2023) provides a suite of six legal reasoning tasks such as rule application and contract understanding, evaluating both correctness and rule-consistency. LexGLUE (Chalkidis et al., 2021) includes classic legal NLP tasks such as case classification, contract QA, and legal entailment, making it suitable for evaluating LLMs in general legal understanding scenarios. LEXTREME (Niklaus et al., 2023) expands this evaluation to 24 languages and 18 tasks, making it a critical multilingual benchmark for assessing FedLLM's cross-lingual legal proficiency in global regulatory contexts.

**(ii) Chinese legal language understanding.** Given the importance of regional legal systems, several benchmarks target Chinese legal capabilities, aligning well with privacy-sensitive applications deployed in Chinese jurisdictions: LawBench (Fei et al., 2023) evaluates Chinese legal LLMs across three cognitive levels—retention, understanding, and application—via 20 legal tasks. LAiW (Dai et al., 2023) offers a fine-grained assessment framework covering fundamental to advanced legal challenges. LexEval (Li et al., 2024b) structures evaluation around a taxonomy of legal cognitive abilities, including memory, reasoning, and application. These benchmarks support federated personalization and regional adaptation of legal agents.

**(iii) Legal citation and generation with formal grounding.** Some legal tasks require not only correct answers but also proper justification grounded in laws and precedents: CitaLaw (Zhang et al., 2024c) focuses on generating legally sound responses with accurate citations to statutes and precedent cases. It uses metrics like syllogism alignment and legal consistency, making it particularly relevant for FedLLM agents operating in jurisdictions with citation and traceability requirements. **(iv) Legal agents and dynamic task-solving.** FedLLM may increasingly support legal assistants or decision-support agents that require multi-step, tool-integrated reasoning: LegalAgentBench (Li et al., 2024a) provides a novel benchmark for evaluating LLM agents on complex, multi-turn legal scenarios. It incorporates intermediate progress tracking and task success scoring, which are useful for evaluating FedLLM performance in decentralized and asynchronous workflows.

**(v) Long-context and multilingual legal reasoning.** Legal documents are often lengthy, hierarchical, and span multiple statutes or precedents: SCALE (Rasiah et al., 2023) is designed to evaluate LLMs on long-context legal documents, legal multilingualism, and cross-document reasoning. Its inclusion of multi-task legal scenarios and code-level legal analysis makes it especially relevant for edge-deployed legal assistants in enterprise or government use cases. Together, these benchmarks provide a rich and diverse framework for evaluating FedLLM in the legal domain—where privacy, jurisdictional customization, long-context understanding, and reasoning fidelity are all critical to real-world deployment. A comprehensive summary of these legal evaluation benchmarks is provided in Table 14.

# 6 Application

## 6.1 FedLLM for Recommendation Systems

Recommendation systems are pivotal across domains such as e-commerce, content streaming, and personalized advertising (Ko et al., 2022). Traditional approaches often rely on centralized data collection, raising significant privacy concerns, particularly when handling sensitive user interactions and preferences. Federated fine-tuning offers a promising alternative by enabling collaborative learning across distributed clients while preserving data privacy.

Zhao et al. (2024a) propose FELLRec, a federated framework for LLM-based recommendation, to tackle the challenges of client performance imbalance and high resource costs. Specifically, FELLRec employs dynamic parameter aggregation and adaptive learning speeds to ensure balanced performance across clients. Additionally, it selectively retains sensitive LLM layers on the client side while offloading other layers to the server, effectively preserving privacy and optimizing resource usage. Similarly, Yuan et al. (2024c) introduce FELLAS, a federated sequential recommendation framework that leverages LLMs as external services to enhance sequential recommendation. FELLAS enriches item embeddings via LLM-assisted textual repre-

sentation while ensuring privacy protection through $d_x$-privacy-compliant sequence perturbation. Beyond privacy protection, FL also facilitates reinforcement learning from human feedback for LLM-based recommendation systems. Wu et al. (2024c) propose FedBis and FedBiscuit, two frameworks designed to enable privacy-preserving federated RLHF. FedBis collaboratively trains a binary selector to filter sensitive preference data, while FedBiscuit clusters clients to train multiple selectors, ensuring better alignment with human preferences while maintaining privacy. Another framework, GPT-FedRec, proposed by Zeng et al. (2024a), integrates ChatGPT with a hybrid Retrieval-Augmented Generation (RAG) mechanism to address data sparsity, heterogeneity, and LLM-specific challenges in federated recommendation. GPT-FedRec employs hybrid retrieval techniques to extract user patterns and item features, then refines recommendations through LLM-generated prompts. By leveraging RAG, the framework effectively mitigates hallucination in LLM-generated content and enhances the overall recommendation quality.

In summary, as LLMs become increasingly integrated into recommendation systems, federated fine-tuning has emerged as a powerful method to fully leverage the capabilities of LLMs while ensuring user data privacy. These advancements demonstrate the potential of FedLLM to support high-quality, privacy-aware recommendation in real-world applications.

## 6.2 FedLLM for Biomedical Research

In the biomedical domain, direct data transmission and centralized model fine-tuning pose significant risks to user and patient privacy. Federated fine-tuning provides a privacy-preserving paradigm that enables collaborative model adaptation across decentralized medical datasets without exposing sensitive information. Ali et al. (2025) explore the use of various FL techniques to fine-tune time-series LLMs on electrocardiogram and impedance cardiography data, enabling privacy-preserving physiological signal analysis. Naseer & Nandakumar conduct a systematic investigation into the application of federated PEFT strategies for fine-tuning vision transformers in medical image classification tasks. Puppala et al. (2024) present a FL-based GPT chatbot designed for personalized healthcare information retrieval. The system aggregates and curates information from diverse sources while ensuring privacy and security through decentralized training. Users receive real-time, personalized insights via an intuitive interface, supported by advanced text parsing, metadata enrichment, and question-answering capabilities. This framework marks a key advancement in patient-centric AI applications.

Sarwar (2025) introduces FedMentalCare, a privacy-preserving framework that integrates FL and LoRA to fine-tune LLMs for mental health analysis. Their study explores the impact of client data volumes and model architectures (e.g., MobileBERT, MiniLM) in FL settings, ensuring scalability, data security, and computational efficiency. Liu et al. (2024c) propose FedFMS, which introduces federated foundation models for medical image segmentation. It addresses privacy challenges in medical imaging by enabling federated training without centralized data sharing. Wang et al. (2024d) introduce FEDKIM, a federated knowledge injection framework for scaling medical foundation models. It leverages lightweight local models to extract private knowledge and integrates it into a centralized model using an adaptive multitask multimodal mixture of experts module, enabling efficient cross-institution knowledge transfer. Dai et al. (2025) propose FedATA, a self-supervised FL framework for medical image segmentation, integrating masked self-distillation with adaptive attention to enhance pre-training and fine-tuning on unlabeled and limited-annotation data. Unlike traditional masked image modeling, FedATA uses latent representations as targets instead of pixels, improving feature learning. Additionally, its adaptive attention aggregation with personalized FL captures institution-specific representations, boosting model generalization and local fine-tuning performance.

In summary, LLMs have become increasingly influential in biomedical research. However, due to the highly sensitive nature of user data in this domain, federated fine-tuning has emerged as a pivotal approach for enabling large-scale AI applications in biomedicine and healthcare while ensuring data privacy and security.

## 6.3 FedLLM for Finance

The financial sector heavily relies on data-driven models for risk assessment, fraud detection, algorithmic trading, and personalized financial services. However, financial data is often highly sensitive, heavily regulated, and distributed across multiple institutions, making centralized model training infeasible due to

privacy concerns and compliance constraints. Federated fine-tuning presents a promising solution by enabling collaborative learning across financial institutions without exposing raw data.

Ye et al. (2024b) introduce OpenFedLLM, a federated fine-tuning framework designed to train LLMs on decentralized private data while ensuring data privacy. In the financial domain, FL-tuned LLMs significantly outperform locally trained models and even surpass GPT-4, demonstrating the potential of FL to enhance LLM performance without compromising sensitive financial data. This study underscores the value of federated fine-tuning in leveraging distributed financial data to develop more accurate, robust, and privacy-preserving LLMs for financial applications. Shabani (2024) explore the use of FL for fine-tuning LLMs in finance, enhancing efficiency and privacy while addressing data scarcity and distribution challenges. Their findings show that FL achieves performance comparable to centralized fine-tuning with significantly lower computational costs and training time, making it ideal for resource-constrained environments. This approach preserves data privacy while enabling the development of more accurate and robust financial LLMs. Similarly, Zeng et al. (2024b) investigate fine-tuning financial LLMs using LoRA and deploying them on edge devices, demonstrating FL's potential to improve both model efficiency and performance in financial applications. Their study highlights significant gains in reasoning capabilities and cost-effectiveness, offering valuable insights into leveraging FL and LLMs for private and vertically specialized financial domains.

In summary, federated fine-tuning plays a crucial role in the financial sector by enabling collaborative model training across institutions while ensuring strict data privacy compliance. This approach allows financial organizations to leverage vast, decentralized datasets for fine-tuning, improving model accuracy without exposing sensitive financial information. As financial markets grow more complex and globally interconnected, FedLLM presents a scalable and secure pathway toward next-generation AI-driven financial infrastructure.

## 7 Open Challenges and Future Directions

**Model Security of FedLLM.** As federated fine-tuning gains momentum, ensuring model security has become a critical concern. In FedLLM, pre-trained models, whether proprietary or open-source, must be transmitted to distributed clients for local fine-tuning, inherently increasing the risk of intellectual property (IP) leakage and system vulnerabilities. Model security in this context involves two key aspects: protecting the IP of high-value models and ensuring the secure deployment of open-source models on edge devices.

First, the financial and strategic value of pre-trained LLMs makes IP protection in FedLLM deployment especially pressing. For example, training models like Gemini Ultra (Mesnard et al., 2024) and GPT-4 (OpenAI, 2023) is estimated to cost $191 million and $78 million, respectively. These models are typically developed by commercial entities under strict licensing and infrastructure control. However, in FedLLM settings, where the full model is often shared with clients in a white-box fashion, it becomes feasible for malicious participants to reverse-engineer or clone the model. This undermines the original developers' competitive advantage and deters participation from commercial model providers. Addressing this challenge requires the development of model watermarking (Pan et al., 2024a), encrypted model delivery, or inference-obfuscation protocols that allow clients to fine-tune and use the model without revealing sensitive architectural or parameter details.

Second, while open-source LLMs (e.g., DeepSeek (Bi et al., 2024), Qwen (Bai et al., 2023a)) are widely adopted in FedLLM due to their accessibility and flexibility, they present new vectors for security threats in federated deployments. In practice, most clients, especially those with limited machine learning or systems expertise, may lack the capabilities to deploy these models securely. For instance, as reported by the technology news outlet AIbase[27], frameworks like Ollama have been found to expose users to data leakage and unauthorized resource usage due to insecure default configurations. In a federated setup, such vulnerabilities are amplified: a single compromised client can leak locally fine-tuned training data, or propagate adversarial backdoors to the global model. The consequences are particularly severe in sensitive domains like healthcare and finance, where breaches may result in the disclosure of protected health information (PHI) or proprietary trading strategies.

To mitigate these risks, future FedLLM research should prioritize the integration of secure model deployment practices into the federated fine-tuning pipeline. Techniques such as confidential computing for secure

---

[27]https://www.aibase.com/news/15909

execution on edge devices, encrypted model delivery, and runtime access control should be incorporated to prevent unauthorized access and tampering. By embedding such security mechanisms into the FedLLM lifecycle, both commercial and open-source models can be safeguarded against misuse, thereby promoting broader adoption in high-stakes domains such as healthcare, finance, and critical infrastructure.

**LLM and SLM Collaboration.** A key future direction for FedLLM lies in enabling efficient collaboration between LLMs and small language models (SLMs) to address the performance–privacy–efficiency trade-offs inherent to federated settings. While LLMs offer superior reasoning and multi-modal capabilities, their large size and resource demands make them impractical for direct deployment on edge devices. Conversely, SLMs such as Gemini Nano (Team et al., 2023) and Phi-3 (Abdin et al., 2024) provide lightweight alternatives with better deployment efficiency but limited generalization and task transferability.

To reconcile these limitations, emerging FedLLM architectures can adopt a hybrid model paradigm: deploying SLMs at the edge for privacy-sensitive inference and lightweight tasks, while offloading complex reasoning or orchestration to cloud-hosted LLMs. This collaborative strategy not only reduces the communication and computation burden at the client side but also enhances regulatory compliance by ensuring sensitive data never leaves the local device. Within this architecture, edge SLMs can perform initial text generation or instruction parsing, while LLMs handle tool selection, global coordination, or cross-domain alignment.

However, realizing this collaboration raises several open challenges in FedLLM: 1) minimizing latency and bandwidth overhead introduced by frequent SLM–LLM interactions; 2) preserving consistency and alignment between local SLM outputs and global LLM behavior; and 3) dynamically adapting task delegation strategies based on client heterogeneity, model confidence, and task complexity. Future FedLLM research should design decentralized orchestration protocols for efficient SLM–LLM coordination, and introduce privacy-preserving metadata exchange mechanisms to protect tool usage logs and inference traces. By enabling seamless SLM–LLM collaboration under federated environments, this hybrid architecture can unlock new levels of scalability, efficiency, and privacy in real-world FedLLM deployments.

**Multi-Modal FedLLM.** While existing FedLLM research has primarily focused on text-based tasks, emerging real-world applications increasingly require multi-modal capabilities, such as integrating visual, speech, and sensor modalities (Zhang et al., 2024a). Large multi-modal models (LMMs), including GPT-4V (Yang et al., 2023b) and LLaVA (Liu et al., 2023b), have demonstrated strong performance in centralized settings. However, extending these models to federated environments presents several unresolved challenges.

A primary difficulty lies in modality heterogeneity across clients (Peng et al., 2024; Ouyang et al., 2023). Devices may possess different input types—for example, some may only have textual data, while others may hold image–text pairs—leading to modality imbalance, where certain modalities are underrepresented in the training process (Fan et al., 2024b). This imbalance can degrade the model's generalization across modalities. Additionally, achieving cross-modal alignment (Gao et al., 2024c)—the model's ability to relate and reason across different modalities—becomes more difficult in federated setups, where paired data (e.g., image–caption pairs) cannot be shared centrally. Moreover, the high computational demands of LMMs further constrain their deployment and fine-tuning on edge devices with limited memory and processing capabilities (Liang et al., 2024; Jin et al., 2024).

To address these challenges, future research should develop modular and flexible tuning frameworks that allow each modality to be fine-tuned independently on client devices. This decoupling enables efficient local adaptation without requiring all modalities to be present on each client. Furthermore, modality-aware aggregation protocols—which weight client contributions based on modality type, data quality, and semantic consistency—can help mitigate imbalance and enhance global model performance. Promising directions also include federated cross-modal contrastive learning (Yu et al., 2023c), which can improve multi-modal alignment without requiring raw data exchange. Finally, to facilitate deployment in edge-centric applications such as smart healthcare, assistive robotics, and wearable systems, it is essential to design lightweight multi-modal architectures through techniques like knowledge distillation (Cai et al., 2024b) or dynamic subnetwork activation (Alam et al., 2022) that strike a balance between accuracy and resource efficiency.

**Continual Learning in FedLLM.**   In dynamic federated environments, client data distributions and task objectives evolve over time, necessitating continual learning capabilities in FedLLM systems (Yoon et al., 2021; Wang et al., 2024e).  Unlike traditional FL settings with fixed tasks and static datasets, real-world deployments require models to incrementally incorporate new knowledge without retraining from scratch. However, continual fine-tuning of LLMs introduces several unique challenges. The sheer size and overparameterization of LLMs make them prone to catastrophic forgetting during incremental updates (Huang et al., 2024a), especially when client participation is sparse or irregular.  Moreover, repeated retraining across rounds is computationally expensive and often infeasible on edge devices with limited hardware resources.

To overcome these limitations, future research should investigate parameter-efficient continual learning strategies that enable local knowledge retention while supporting scalable global updates.  Techniques such as Elastic Weight Consolidation (EWC) (Kirkpatrick et al., 2017), PEFT-based modular updates, and rehearsal methods using compressed memory buffers (Tiwari et al., 2022) are promising in mitigating forgetting without incurring prohibitive overhead.  In addition, the design of lifelong personalization protocols—capable of adapting to each client's evolving task distribution under Non-IID and intermittent data availability—remains an open research frontier. Developing such protocols requires balancing communication efficiency, privacy preservation, and model stability across heterogeneous learning trajectories.

Ultimately, enabling continual adaptation in FedLLM will be essential for long-term deployment in dynamic real-world scenarios, such as personalized healthcare, evolving legal compliance systems, or lifelong learning assistants.  This calls for a shift from round-based static fine-tuning to streaming, task-aware federated adaptation frameworks that can incrementally evolve with users and environments.

**Memory-Efficient FedLLM.**   Memory efficiency remains one of the most fundamental and restrictive bottlenecks in the deployment of FedLLM—often resulting in a binary feasibility condition: a client device either meets the memory requirements to participate in training or is entirely excluded (Wu et al., 2025d). Unlike other challenges such as communication overhead or data heterogeneity, which degrade performance but still permit participation, memory limitations can preclude participation altogether, particularly for edge devices with constrained hardware capabilities. Although PEFT techniques like LoRA substantially reduce the number of trainable parameters, they fall short of fully mitigating memory pressure. For example, fine-tuning LLaMA2-13B with LoRA still demands over 50 GB of peak memory—an order of magnitude beyond the capacity of most mobile phones, IoT devices, or embedded systems (Xu et al., 2024b; Tian et al., 2024a).

Addressing this limitation requires innovation at both the algorithmic and system levels. On the algorithmic side, emerging approaches such as dynamic layer-wise adaptation (Pan et al., 2024b), quantization-aware PEFT (e.g., QLoRA) (Dettmers et al., 2023), and structured model pruning (Wang et al., 2019b; Ma et al., 2023) offer promising pathways for reducing memory footprints during local fine-tuning.  Complementing these are system-level solutions such as gradient checkpointing and accumulation (Gim & Ko, 2022), runtime memory-aware schedulers, and cloud-edge hybrid training architectures with selective computation offloading (Kumar et al., 2013), all of which aim to stretch the effective memory capacity of participating devices. To fully unlock the potential of FedLLM at scale, future research should explore co-designed frameworks that jointly optimize algorithmic efficiency and system-level deployment. Such holistic solutions can harmonize memory, computation, and communication trade-offs in real time—enabling resource-adaptive, privacy-preserving model customization across highly diverse client ecosystems.

Overcoming the memory barrier would expand the pool of eligible participants to include billions of low-memory edge devices that are currently sidelined from federated training.  This not only enhances the inclusiveness, representativeness, and scalability of the FedLLM framework, but also opens the door to real-world deployments in settings such as home automation, wearables, and low-power industrial IoT platforms.

# 8   Conclusion

To the best of our knowledge, this is the *first* comprehensive survey dedicated to the federated fine-tuning of LLMs.  We begin by introducing foundational background knowledge and identifying four core challenges through empirical analysis, which reveal the fundamental limitations that federated fine-tuning must overcome.  We then review the latest relevant research papers, systematically organizing recent advances

in parameter-efficient federated fine-tuning techniques. These approaches are categorized based on their methodologies, with detailed discussions on how each class of methods addresses the identified challenges. Furthermore, we present a comprehensive evaluation framework encompassing both fine-tuning datasets and evaluation benchmarks across different domains, offering a holistic framework for assessing FedLLM performance. Beyond methodological contributions, we highlight practical applications of FedLLM across domains. Finally, we outline promising future directions in this rapidly evolving field. While notable progress has been achieved, several pressing challenges remain open. Addressing these issues will be essential to unlocking the full potential of FedLLM and enabling its widespread deployment in practical, privacy-sensitive applications.

## Acknowledgements

This work is supported in part by the Science and Technology Development Fund of Macau (0107/2024/RIA2), Joint Science and Technology Research Project with Hong Kong and Macau in Key Areas of Nansha District's Science and Technology Plan (EF2024-00180-IOTSC), and the Multi-Year Research Grant of University of Macau (MYRG-GRG2023-00211-IOTSC-UMDF and MYRG-GRG2024-00180-IOTSC).

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
