# OpenReview forum: "A Survey on Federated Fine-Tuning of Large Language Models"
_TMLR — Accepted by TMLR_

### Review · Reviewer_Tc4r · 2025-09-01

**Summary Of Contributions:**

The paper surveys federated fine-tuning of LLMs, namely “FedLLM', covering challenges through the lens of communication, heterogeneity, memory, compute perspectives. In addition to the survey, the authors also introduce a PEFT-centric taxonomy (Parameter-Efficient Fine-Tuning) for federated settings, datasets/benchmarks across domains, applications (finance/biomed/recsys), and other open problems. The authors go through a thorough review of existing fine-tuning datasets and evaluation benchmarks and rigorously evaluate the performance of FedLLM. Moreover, the authors provide a comprehensive discussion in terms of the real-world applications across multiple domains, as well as identify critical open challenges and outline promising research directions to foster future advancements in FedLLM.

**Audience:**

Yes

**Audience Explanation:**

The paper conducts sufficient and different domains that are related to FedLLM, including method, potential applications and future open problems (more details are in the above section). These contents could be helpful for readers who want to dive deep into these directions.

**Broader Impact Concerns:**

No significant broader impact concerns.

**Claims And Evidence:**

Yes

**Claims Explanation:**

The paper focuses on federated fine-tuning and motivates the need with concrete resource barriers and privacy considerations, plus empirical memory/FLOP profiles, providing a clear scope and motivation.

The paper includes and contrasts related surveys along privacy/LLM/efficiency/benchmarks/applications/future work, and summarize these comparisons in tables, which helps readers see the niche.

Beyond the standard PEFT categories, the paper proposes a LoRA-specific breakdown (homogeneous / heterogeneous / personalized) with representative methods and aggregation ideas—helpful in practice.

The general-domain and domain-specific benchmark tables are extensive (finance, medical, code, long-context), and many entries are annotated with evaluation criteria—useful for experiment planning.

Sections on finance, biomed, recsys, and future directions (e.g., IP protection, SLM–LLM collaboration, memory-efficient FedLLM) make the survey actionable.

**Requested Changes:**

The survey has done sufficent works in exhaustive review, but I would suggest providing links/hyperlinks in the paper when summarizing the datasets and benchmarks, to make sure the readers get the same resources referenced in the paper.

The references contain non-scholarly resources, .e.g, news item from AIbase about Ollama vulnerabilities. It would be better to justify the correctness of these resources and prioritize the usage of peer-reviewed sources. Otherwise, we could mark it to ask the readers to be careful about this resource.

The settings of the evaluated performance are not referenced or provided. Figure 7 lacks enough detail, such as hardware, precision.

Some citations are duplicated, e.g. GSM8K.

---

### Review · Reviewer_UmEG · 2025-09-18

**Summary Of Contributions:**

The paper surveys federated fine-tuning of LLMs (FedLLM). It (1) outlines background on LLM training and why centralized and local fine-tuning fall short, then (2) analyzes four practical bottlenecks—communication, data heterogeneity, memory, and compute. It (3) organizes parameter-efficient strategies for FedLLM, including LoRA, prompts, adapters, and selective tuning, with sub-taxonomies (e.g., homogeneous/heterogeneous/personalized LoRA) and representative methods. It (4) catalogs datasets and benchmarks and (5) sketches applications (finance, biomedical, recommendation). Finally, it (6) flags open directions such as continual learning and memory-efficient FedLLM as critical for real-world deployment.

**Additional Comments:**

None

**Audience:**

Yes

**Audience Explanation:**

This paper provides a thorough survey of PEFT in federated LLMs, summarizing core methods and benchmarks and serving as a good starting point for newcomers to FedLLMs fine-tuning.

**Broader Impact Concerns:**

No concerns.

**Claims And Evidence:**

Yes

**Claims Explanation:**

There is a coherent taxonomy of FedLLM methods (LoRA / prompts / adapters / selective), laid out with an overview figure and dedicated subsections. This is clearly evidenced in the structure and visuals (Fig. 2 & Fig. 9) and the LoRA/prompt preliminaries.

Compute constraints are substantiated by a profiling exercise: the paper presents FLOPs comparisons across model sizes and argues why this impedes on-device training (with citations). The evidence exists as text + figure and is reasonably clear.

Applications coverage (finance/biomed/recsys) is present and organized, which supports the claim that FedLLM is relevant beyond toy benchmarks.

**Requested Changes:**

The survey is comprehensive overall: it succinctly summarizes current PEFT methods for FedLLMs, identifies key challenges and applications, and outlines meaningful directions for future research. I have only a few minor revision requests.

1. Figure 1 panel (a): "Break data privay" → Break data privacy.

2. Intro: "computation on on 2,048 NVIDIA A100 GPUs" → remove duplicate "on."

3. "SA-FedLora" $\rightarrow$ "SA-FedLoRA"

---

### Review · Reviewer_bgda · 2025-10-20

**Summary Of Contributions:**

This paper comprehensively summarizes the federated fine-tuning of LLMs. It analyzes four main challenges (communication overhead, data heterogeneity, memory constraints, and computation burden) in FedLLM and provides five different federated fine-tuning methods. Different from current existing surveys, it’s the first survey to organize the evaluation datasets and benchmarks for federated fine-tuning of LLMs. Furthermore, it proposes five opening research directions to contribute future federated fine-tuning of LLMs.

**Audience:**

Yes

**Audience Explanation:**

The paper is well-written and easy to follow, and it's a comprehensive survey including and organizing almost all existing works in federated fine-tuning of LLMs.

**Broader Impact Concerns:**

No.

**Claims And Evidence:**

Yes

**Claims Explanation:**

The paper covers almost all existing works in federated fine-tuning of LLMs, and clearly organize them into different categories. For example, for LoRA in federated fine-tuning, it utilizes three scenarios (homogeneous LoRA, heterogeneous LoRA, personalized LoRA) to conclude the works.

**Requested Changes:**

1. The text size in Table 9, Table 10, and Table 11 is too small for readers.

2. In the Introduction section, the paper organizes LLM fine-tuning paradigms into three approachs, which are central fine-tuning, local fine-tuning, and federated fine-tuning. It’s not clear to me about the advantages of federated fine-tuning, compared to local fine-tuning. Does the global model trained on all clients have the same good performance with the local fine-tuned model? Since there may have some disalignment when merging all clients weights, which may affect the performance of the global model on some specific clients. Can authors explain this clearly? Also, the goal of some federated fine-tuning methods is not to have one single global model but to have several global models. The Figure 1(c) may consider this point.

---

> ### Author Response · Authors · 2025-10-30
>
> Thank you for your thorough review and positive feedback on our paper. We are encouraged that you found our survey to be comprehensive, well-written, and a valuable contribution to the TMLR audience. Following your suggestions, we have diligently revised the manuscript. Below, we detail the changes made in response to each of your points.
>
> > __Q1: Text Size in Tables.__
>
> We appreciate the reviewer's feedback. We agree that readability is essential. To **resolve this issue**, the revised manuscript features **completely restructured** versions of Tables 9, 10, and 11. We have split these tables and refined their presentation to **guarantee** legibility and clarity for all readers.
>
>
>
> > **Q2: Clarification on Federated Fine-tuning vs. Local Fine-tuning.**
>
> Thank you for this insightful question, which highlights the central trade-off between local specialization and global generalization. You are correct that naively merging client weights can lead to performance degradation due to **data heterogeneity**, a key challenge we discuss in Section 3.2.
>
> First, we wish to clarify that the primary advantage of federated fine-tuning is its ability to build a **powerful global** model by leveraging diverse, decentralized data while **preserving data privacy**. The global model generally **outperforms** one trained only on local data. This is because local fine-tuning, while perfectly adapted to a single client's data, often suffers from poor generalization and suboptimal performance when the local data is limited in scale or diversity. As we note in our introduction, this can lead to significant performance drops (e.g., **up to 7%** on the MMLU benchmark).
>
> Second, regarding your concern about **weight misalignment**, many of the advanced techniques surveyed in our paper (e.g., regularization-based methods and modified aggregation strategies, discussed in Sections 3.2 and 4) are **specifically designed to mitigate this "client drift."** These methods aim to produce a robust global model that performs well for all participants. Therefore, while the resulting global model may not be a perfect match for every client's local data distribution, it typically achieves superior overall performance and generalization by leveraging the breadth of the collective data.
>
>
>
> > **Q3: Figure 1(c) & Multi-Global-Model Paradigm.**
>
> We thank the reviewer for this insightful question, which cuts to the core of federated learning. Our intention with Figure 1(c) was to illustrate the **foundational** federated process—the classic paradigm where local models are aggregated into a **single** global model.
>
> The reviewer is **entirely correct** in pointing out that paradigms involving **multiple global models** exist. These, in fact, **directly correspond** to the advanced personalization techniques we survey in Section 3.2. Such methods indeed produce multiple tailored models to better adapt to client data heterogeneity. Even in these sophisticated, personalized scenarios, the models still **rely on and benefit immensely** from the collective knowledge shared across the system. This detail actually **underscores** the fundamental power and flexibility of the federated paradigm and its definitive advantage over isolated, local-only fine-tuning.
>
> In the revised version, we update the caption for Figure 1(c) to clarify that it depicts this classic aggregation process. Furthermore, we expand our discussion of personalization in Section 3.2 to explicitly highlight these multi-global-model approaches.
>
> Thank you once again for your valuable and constructive feedback. We hope these adjustments will effectively address your concerns.

---

### Decision · Action_Editor_vNsE · 2026-01-24

**Recommendation:** Accept as is

**Audience:**

Yes

**Audience Explanation:**

Yes. This paper would be of interest to a substantial portion of the TMLR audience, particularly researchers working on federated learning, large language models, and efficient optimization. The survey provides a timely and comprehensive overview of federated fine-tuning methods, challenges, benchmarks, and applications, making it a valuable reference for both newcomers and researchers seeking a structured understanding of this rapidly evolving area.

**Claims And Evidence:**

Yes

**Claims Explanation:**

The submission’s claims are supported by clear and convincing evidence appropriate for a survey paper. The authors systematically organize and synthesize the existing literature on federated fine-tuning of LLMs, provide well-structured taxonomies with representative citations, and substantiate key challenges (e.g., communication, heterogeneity, and compute constraints) using concrete examples and profiling results.